# IMAGE QUALITY ASSESSMENT FOR EMBODIED AI

**Chunyi Li**[1,2,3,*] **Jiahao Xiao**[1,2,*] **Jianbo Zhang**[1,*] **Farong Wen**[1,2] **Zicheng Zhang**[2] **Yuan Tian**[2]
**Xiangyang Zhu**[2] **Xiaohong Liu**[1] **Zhengxue Cheng**[1,†] **Weisi Lin**[3] **Guangtao Zhai**[1,2,†]
[1] Shanghai Jiao Tong University   [2] Shanghai AI Lab   [3] Nanyang Technological University
Project Page: *https://github.com/aiben-ch/EmbodiedIQA*

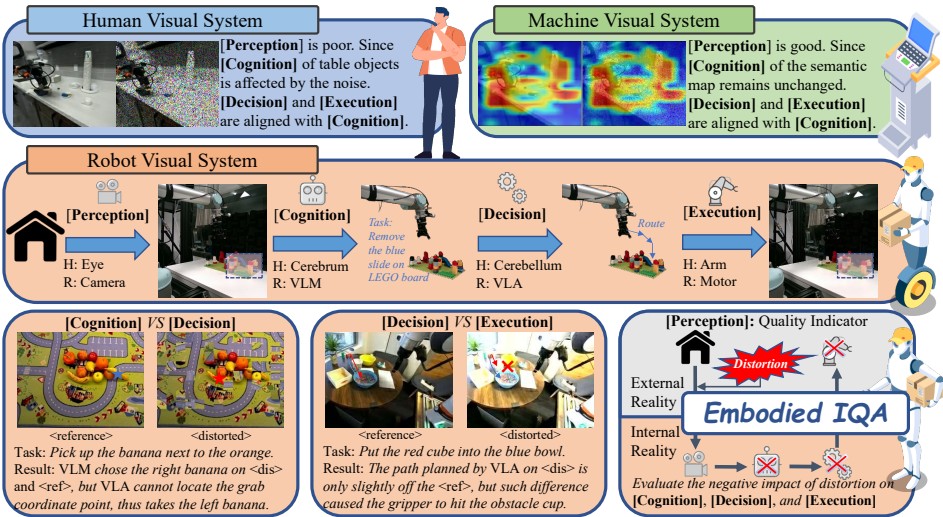

Figure 1: The significant gap between human, machine, and robot visual systems. Humans and Machines are sensitive to different distortions, while Robots have **Decision** and **Execution** steps beyond **Cognition**, highlighting the importance of a **Perception** quality index for Embodied AI.

## ABSTRACT

Embodied AI has developed rapidly in recent years, but it is still mainly deployed in laboratories, with various distortions in the Real-world limiting its application. Traditionally, Image Quality Assessment (IQA) methods are applied to predict human preferences for distorted images; however, there is no IQA method to assess the usability of an image in embodied tasks, namely, the perceptual quality for robots. To provide accurate and reliable quality indicators for future embodied scenarios, we first propose the topic: IQA for Embodied AI. Specifically, we (1) based on the Mertonian system and meta-cognitive theory, constructed a perception-cognition-decision-execution pipeline and defined a comprehensive subjective score collection process; (2) established the Embodied-IQA database, containing over 30k reference/distorted image pairs, with more than 5m fine-grained annotations provided by Vision Language Models/Vision Language Action-models/Real-world robots; (3) trained and validated the performance of mainstream IQA methods on Embodied-IQA, demonstrating the need to develop more accurate quality indicators for Embodied AI. We sincerely hope that through evaluation, we can promote the application of Embodied AI under complex distortions in the Real-world.

## 1 INTRODUCTION

To achieve Artificial General Intelligence (AGI), Embodied AI, as a bridge connecting external and internal realities, has developed rapidly in recent years. Relying on its ability to interact with the physical environment, Embodied AI Duan et al. (2022); Savva et al. (2019); Liu et al. (2024b) has been applied to simple scenarios such as factories and warehouses, but it is not yet capable of handling complex environments like autonomous driving and wilderness exploration. Unlike traditional robotics driven by fixed algorithms, Embodied AI collects signals from the Real-world

and is therefore susceptible to distortions. For example, a pick-and-place task may be successfully debugged in the laboratory, but it may fail in Real-world applications due to slight lens shaking. Therefore, the preferences of Embodied AI should be analyzed to filter out these low-quality images.

For human viewers, this problem can be solved through Image Quality Assessment (IQA) metrics. For example, in streaming media, collect human subjective preferences for distorted images. Since human resources are expensive, IQA will develop objective quality indicators to fit subjective scores. Similarly, for Embodied AI, it is also necessary to collect the success rate of downstream tasks using distorted images, quantifying the fidelity with the reference results, and developing IQA metrics.

Unfortunately, due to the significant differences between Human/Machine/Robot Visual Systems (HVS/MVS/RVS), previous IQA methods cannot be directly transferred to Embodied AI scenarios, as shown in Figure 1. First, HVS and MVS are sensitive to different types of distortions. Humans are sensitive to distortions such as noise and compression, which do not affect the downstream tasks of machines. Brightness and contrast are the opposite. Therefore, as a machine, Embodied AI cannot use the past human-oriented processing methods. Second, although MVS and RVS are sensitive to some common distortions, the perceptual quality of a general machine only depends on the performance of segmentation and detection tasks that belong to Cognition. Robots, however, have subsequent Decision and Execution steps. High fidelity in the previous step does not guarantee the success of the next. Therefore, unlike a general machine, it is necessary to fully consider Cognition, Decision, and Execution to characterize the Perception of Embodied AI. Considering these issues, we first attempt to implement IQA metrics into Embodied AI. Our contributions are summarized as follows:

- Theory: We refer to the Mertonian Law in robotic intelligence to construct the Perception-Cognition-Decision-Execution pipeline. We define tasks related to Embodied Perception and specify the subjects for each step in Cognition, Decision, and Execution.

- Data: We add corruption to images in Embodied tasks, collecting over 36k reference/distorted image pairs. We perform inference using Vision Language Models (VLM) and Vision Language Action-model (VLA) for over 5 million annotations. This large-scale database can effectively drive the development of quality metrics for Embodied AI.

- Experiment: We experiment with 15 advanced IQA methods on our database, proving that more sophisticated IQA metrics are needed for Embodied AI. Additionally, we first conduct real-world experiments in the IQA field, executing 1.5k Embodied tasks in the Real-world, revealing the internal connections between Cognition, Decision, and Execution.

## 2 RELATED WORKS

### 2.1 MERTONIAN SYSTEM FOR ROBOTIC INTELLIGENCE

Intelligent models are divided into Newtonian and Mertonian Wang (2012) systems. Newtonian systems typically refer to those systems that can be precisely described and predicted by deterministic physical laws, such as classical mechanical systems. In contrast, Mertonian systems involve systems that include 'free will', whose behavior is influenced by feedback between beliefs and actions. The characteristic of such systems is that even given the current state and control conditions, the next state of the system cannot be accurately obtained by solving, so its behavior is difficult to predict precisely.

HVS and MVS can both be simplified as Newtonian systems, since their Decision and Execution processes are robust. However, the Decision and Execution of RVS do not fully match Cognition. For example, a deviation of one character in Cognition may greatly change the pose in Decision; a one-centimeter path offset in Decision may also cause Execution to hit obstacles. Since the impact of distortion on these steps is unpredictable, it is necessary to handle them separately for the IQA task.

### 2.2 IMAGE QUALITY ASSESSMENT FOR MACHINE

Since 1999, Perception has been recognized as the first step in the interaction between AI agents and external reality, whose mechanism Rickel & Johnson (1999); Cassimatis et al. (2004); Lepora & Pezzulo (2015); Balke & Gilbert (2014) has been revealed. However, no perceptual quality score has been assigned to each distorted image like current IQA Li et al. (2024b; 2025b); Liu et al. (2024a) metrics, which is exactly Embodied AI needs in Real-world applications. In the past decades, IQA has been widely studied as shown in Table 1, but none of them meet the above needs of Embodied AI.

Table 1: Comparison of Embodied-IQA with other perceptual quality databases. As a machine-oriented database, Embodied-IQA has not only more image samples and a larger annotation scale, but also comprehensive labels on three downstream steps. [Keys: Cognition, Decision, Execution]

| Database | Image | | | Corruption | | Annotation | | | Cog. | Dec. | Exe. |
| | Reference | Distorted | Resolution | Types | Strength | Num | Dimension | Subjects | | | |
|---|---|---|---|---|---|---|---|---|---|---|---|
| LIVEMoorthy & Bovik (2011) | 29 | 779 | 768 | 5 | 5 | 25k | 1 | Human (General) | ☑ | | |
| TID2013Ponomarenko et al. (2015) | 25 | 3k | 512 | 24 | 5 | 514k | 1 | Human (General) | ☑ | | |
| KADID-10KLin et al. (2019) | 81 | 10k | 1k | 25 | 5 | 304k | 1 | Human (General) | ☑ | | |
| CLIC2021Ballé & et al. (2020) | 585 | 3k | 1k | 10 | 3 | 484k | 1 | Human (General) | ☑ | | |
| NTIRE2022Gu et al. (2022) | 250 | 29k | 288 | 40 | 5 | 1.13m | 1 | Human (General) | ☑ | | |
| AGIQA-3KLi et al. (2023) | - | 3k | 1k | - | - | 125k | 1+1 | Human (Multimodal) | ☑ | ☑ | |
| NTIRE2024Li et al. (2024a) | - | 20k | 1k | - | - | 420k | 1+1 | Human (Multimodal) | ☑ | ☑ | |
| MPD Li et al. (2025a) | 1k | 30k | 1k | 30 | 5 | 2.25m | 5 | Machine (General) | ☑ | | |
| EPD Zhang et al. (2024a) | 100 | 2.5k | 256 | 25 | 5 | 30k | 2 | Machine (General) | | ☑ | |
| **Embodied-IQA (ours)** | **1.23k** | **36.9k** | **1k** | **30** | **5** | **5.53m** | **3+3+1** | **Machine (Robot)** | ☑ | ☑ | ☑ |

First, most databases are human-oriented, with only two coarse-grained Zhang et al. (2024c;d), two fine-grained Li et al. (2025a); Zhang et al. (2024a) machines as subjects (VLM and reinforcement learning); second, as mentioned above, no database covers Cognition, Decision, and Execution altogether. Considering the characteristics of Embodied AI, it is necessary to establish a new dataset in accordance with the requirements of the Mertonian system.

# 3 DATABASE CONSTRUCTION

## 3.1 REFERENCE & DISTORTED IMAGE COLLECTION

To comprehensively characterize the data in Embodied scenarios, we collect 1,230 high-quality samples as reference images. (see Supplementary for data source) All data are pre-processed by Q-Align Wu et al. (2024) to avoid pre-distortion before adding distortion. We focused on two aspects, Sim2Real and Perspective, to ensure coverage of both real and simulation, as well as first-person and third-person perspectives. In addition, we divided the subjects and backgrounds into five categories each, as shown in Figure 2, to ensure versatility by involving each category in the database.

For the distorted images, according to the corruption caused by the current communication protocols, 30 distortion types are considered and classified into 7 categories: Blur, various types of unclear image; Luminance, global brightness changes; Chrominance, global color changes; Noise, random noise of different distributions; Compression, codec algorithm like JPEG; Spatial, local pixel-level changes; and others. For each distortion, we defined 5 intensity levels, ensuring the quality degradation perceived by the HVS is aligned at the same level. Thus, for each reference image, we randomly selected the intensity to add all the corruptions mentioned above, resulting in 36,900 distorted images. The reference/distorted image pairs will then be annotated by Embodied AI subjects.

## 3.2 PERCEPTION: TASK DEFINITION

Perception refers to receiving information about the external environment, where Embodied AI obtains information through sensors, similar to human sensory organs. Considering that more than 82% of human external input signals come from vision, we simplify this step of Embodied AI to the camera. First of all, we need to clarify the factors that Embodied AI focuses on in Perception. When viewing an image, HVS pays attention to factors such as brightness/chromaticity, while MVS/RVS relies on specific downstream tasks. Therefore, based on information such as objects, layout, and environment in the image, we manually annotate 5 tasks for each reference sample in natural language, as in previous MVS Li et al. (2025a) works. The difficulty of the tasks here increases in sequence and is limited to [`Cover`, `Insert`, `Move`, `Pick`, `Place`, `Pour`, `Press`, `Pull`, `Push`, `Twist`] to avoid being too difficult. All subsequent steps are based on the task corresponding to each image, and the image quality depends on the similarity of the reference/distorted image pair inference results.

## 3.3 COGNITION: VLM ANNOTATION

Cognition refers to the process of processing and understanding information after perception, including recognition, classification, memory, and reasoning. This function of Embodied AI is implemented

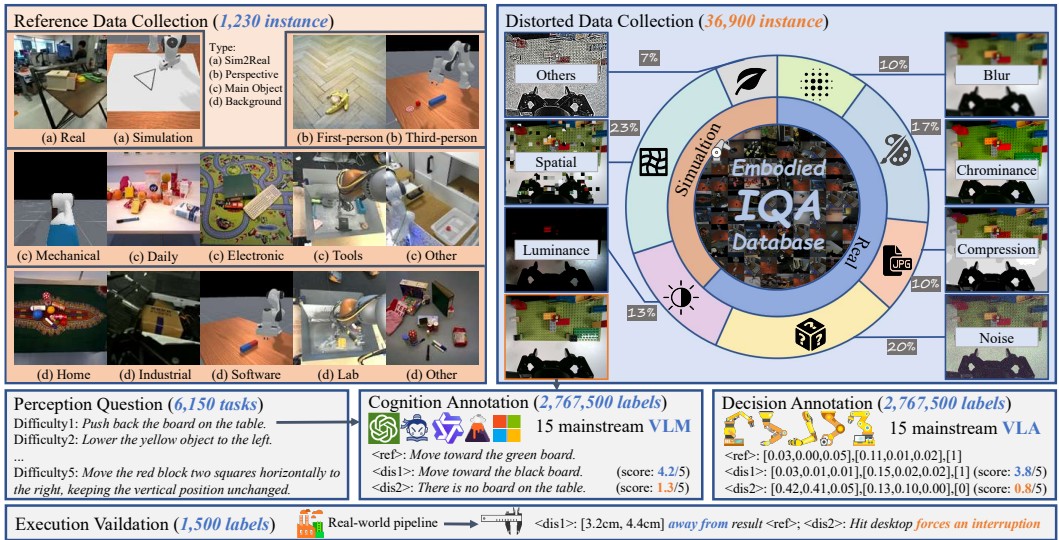

Figure 2: Database construction of the Embodied-IQA, with 30k+ large-scale reference/distorted image pairs, meticulously annotated with 2m+ fine-grained **Cognition** score from 15 mainstream VLMs, 2m+ **Decision** score from 15 VLAs, and 1.5k real-world experiments as **Execution** score.

through VLM, which corresponds to the human cerebrum. Specifically, we selected 15 commonly used VLMs for cognition, including: Mini-InternVL Gao et al. (2024), InternLM-Xcomposer2 Dong et al. (2024), InternLM-Xcomposer2.5 Zhang et al. (2024b), InternVL2 Chen et al. (2024b), InternVL2.5 Chen et al. (2025), InternVL3 Zhu et al. (2025), MPlugOwl3 Ye et al. (2024), Ovis1.5-Gemma Team et al. (2025), Ovis1.6-Llama Touvron et al. (2023), Ovis2 Lu et al. (2024), Phi3-Vision Abdin et al. (2024b), Phi3.5-Vision Abdin et al. (2024a), Phi4-Multimodal Microsoft et al. (2025), Qwen2-VL Yang et al. (2024), and Qwen2.5-VL Bai et al. (2025). To ensure usability in the Real-world, the parameter size we selected is all below 8B to ensure real-time inference.

Considering the output modality is textual, VLMs will be required to solve the pre-defined task in about 10 words, and the difference between reference/distorted output sentences will be measured. Specifically, the difference between the two output sentences includes three dimensions: accuracy, recall, and semantics, which are realized by the average of three classic indicators: BLEU Papineni et al. (2002), ROUGE Vedantam et al. (2015), and CIDEr Vedantam et al. (2015).

### 3.4 DECISION: VLA ANNOTATION

Decision refers to selecting the best course of action based on goals, rules, and experience, according to Cognition results. This function of Embodied AI is implemented through VLA, which corresponds to the human cerebellum. Specifically, we selected 15 commonly used VLA for Decision, including: CogACT Li et al. (2024c), Embodied-CoT Zawalski et al. (2025), Octo Team et al. (2024), OpenVLA Kim et al. (2024a), OpenVLA-Libero Kim et al. (2024c), OpenVLA-Goal Kim et al. (2024b), OpenVLA-Libero-Object Kim et al. (2024b), OpenVLA-Libero-Spatial Kim et al. (2024b), Pi0-Aloha-Pen Black et al. (2024b), Pi0-Aloha-Towel Black et al. (2024b), Pi0-Aloha-Tupperware Black et al. (2024b), Pi0-Base Black et al. (2024a), Pi0-Droid Pertsch et al. (2025), Pi0-Fast Pertsch et al. (2025), and RT-X-1 Collaboration et al. (2025), whose parameter size is controlled at 8B.

Noted that since Embodied-IQA first introduced VLA into the IQA task, we define the quality of VLA as three dimensions. First, we parse the 7-DoF Pose[1] output field. According to the mechanism of VLA, the first three represent position [2] (translation of the operator along the three-dimensional coordinate system, in mm), the middle three represent rotation (rotation of the operator along the three-dimensional coordinate system, in rad), and the last one represents state (opening and closing of the operator, range [0-1]). The position score is based on the spatial distance of the coordinate points

---

[1]We will discard information beyond the above 7-DoF like depth, for alignment between VLAs.

[2]For two-arm VLAs, we only select the arm with the larger movement range.

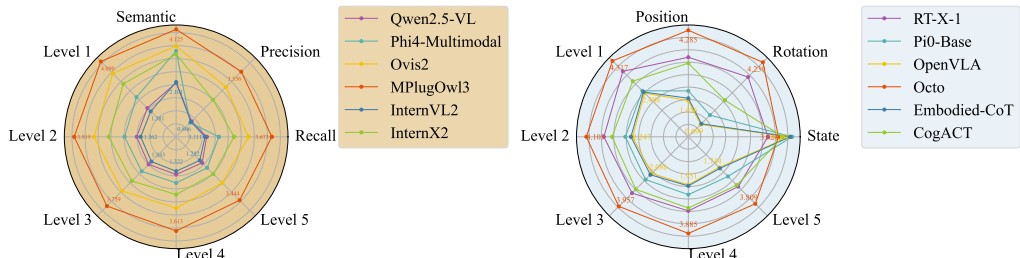

Figure 3: Benchmarking **VLMs**&**VLAs** in 3 different score dimensions and 5 distortion levels. Their performance varies in 3 dimensions and decreases with the distortion. (Zoom in for detail)

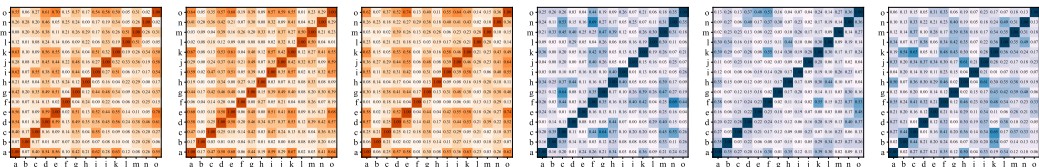

Precision (0.31)   Recall (0.31)   Semantic (0.30)   Position (0.25)   Rotation (0.23)   State (0.28)

Figure 4: Correlation matrix of **VLMs**&**VLAs** subjects, the a-o order follows Section 3.3,3.4. Darker colors denote a higher SRCC, with the averaged SRCC attached to the bottom of the matrix.

obtained from the reference/distorted image, the rotation score is based on the cosine similarity of directional vectors, and the last one is the absolute difference. The three dimensions are averaged after 0-1 normalization, and report the sum of the five task results as the final Decision score.

## 3.5 EXECUTION: REAL-WORLD VALIDATION

Execution refers to the process of transforming decisions into actual movements. This part of Embodied AI relies on specific actuators, corresponding to the human motor system. Based on kinematic statistics, the upper limbs dominate among all muscles and complete over 50% of daily movements. Therefore, we use the robotic arm as the most representative actuator. Specifically, we execute tasks based on the inference results of VLA, with three scenarios: (1) Success: Directly assign score 100 to the sample; (2) Failure: Measure the Euclidean distance between the reference and distorted results based on the final pose of the actuator, and deduct points in centimeters; (3) Emergency stop: If the actuator hits the table or wall, directly assign score 0. Considering the uncontrollable factors in real-machine experiments, we only execute the task with the lowest difficulty level among the 5 tasks to verify whether the results of VLM and VLA align with the Real-world.

## 4 DATABASE ANALYSIS

This section analyzes Embodied-IQA database from four dimensions: On the model level, we (1) Benchmark the VLMs and VLAs when processing the distorted images; (2) Explore the internal correlation between VLMs and VLAs; On the instance level, we (3) Compare the score distributions under different categories and distortions; (4) Analyze the distortion sensitivity of VLMs/VLAs.

[Benchmark] We select 6 representative VLM and VLA in Figure 3. As the distortion level increases, the total scores of both VLM and VLA gradually decrease. However, the differences among the three scoring dimensions of VLM are much greater than the level of distortion. After distortion, the Semantic score of the image decreases relatively little, followed by Recall, and then Precision. In VLM, the reference/distorted output of MPlugOwl3 is the most consistent, while advanced models like Qwen2.5-VL are less robust. Therefore, distortion usually affects VLM at the character level rather than the semantic level, and it is more likely to output redundant text than to lose information. Meanwhile, there are also significant differences among the three scoring dimensions of VLA. In VLA, Octo shows strong robustness to distortion in Position and Rotation, while models like CogACT and OpenVLA are more faithful in State. Among them, State changes little after distortion, Position changes more, and Rotation is the most easily affected by distortion. This indicates that distortion

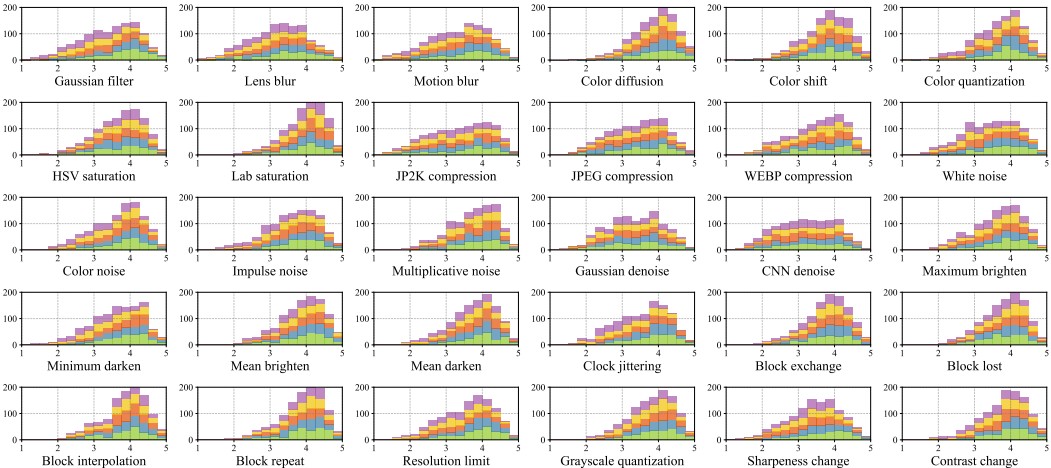

Figure 5: Decision score visualized in 30 distortion subsets. Different color denotes distortion **Level 1**-**Level 2**-**Level 3**-**Level 4**-**Level 5**. Different distortions affecting VLAs vary significantly.

usually does not affect the end operator but has a significant impact on the robot arm. Therefore, VLA needs to be carefully selected to ensure high-quality output of the first 6-DoF.

**[Correlation]** We visualize the correlation between subject models in Figure 4 according to the order of models in Section 3.3,3.4 through Spearman Rank-order Correlation Coefficient (SRCC). According to past IQA Li et al. (2025a) work, the correlation of HVS is usually above 0.6, while that of MVS is often less than 0.5. In specific machine tasks, detection has better correlation, while Visual Question Answering (VQA) may even be less than 0.4. For Cognition, which uses VLM to solve embodied tasks (a degraded form of VQA), the correlation is only about 0.3. Moreover, we find that the correlation of VLA is even lower than that of VLM, at around 0.25. Therefore, when evaluating VLM and VLA, using only one model as the subject is far from sufficient, especially for VLA. It is necessary to collect their general preference. This also reflects the necessity of constructing the Embodied-IQA database and the separation of Cognition and Decision in the RVS.

**[Distribution]** Since Decision is more downstream than Cognition and has never been deeply investigated in IQA, we show the distribution of Decision and put Cognition in the supplementary. Figure 5 shows the Decision score distribution under 30 types of distortions and 5 intensity levels. Results show that RVS and the traditional HVS have significant differences. Taking brightness as an example, Embodied AI is highly sensitive to 'Maximum brighten/darken', and the quality significantly decreases with the distortion level; however, it is rarely affected by 'Mean brighten/darken', and there is no significant distribution change from level 1 to 5. These findings emphasize the differences between RVS and HVS. Figure 6 lists the relation of the three Decision dimensions and the score distributions corresponding to different image categories. Overall, Position, Rotation, and State show independent distributions. In the Sim2Real distribution, the real scores are relatively high, indicating that VLA is better at Real-World data; the first-person results are far worse than the third-person results, indicating that in the training data of VLA, the sampling tools and actuators are rarely integrated, which needs to be improved in the future. These findings jointly support the rationality of the division of source image data and annotation dimensions in Embodied-IQA.

**[Sensitivity]** The difference between MVS and HVS causes VLM to be highly sensitive to some distortion categories, but robust to others; the difference between RVS and HVS also causes VLA to have a similar phenomenon. We combined the Just-Noticeable-Difference (JND) theory to analyze the similarities and differences in the distortion sensitivity of VLM and VLA, as shown in Figure 7. The sensitivity is divided into three levels, Mild, Medium, and Severe, each accounting for one-third of all image samples, according to the Cognition and Decision scores. Results shows although VLM and VLA have certain commonalities in sensitivity, there are also distortions such as Dis02 'Lens blur' that mainly affect VLM, or Dis15 (Multiplicative noise) that mainly affect VLA. This VLM&VLA-based partition further explains the gap between Cognition and Decision, and can serve as an important reference in the IQA for the Embodied AI topic in the future.

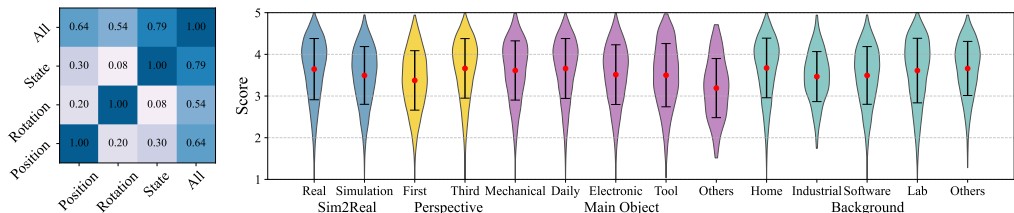

Figure 6: Correlation between the general Decision score and the 3 dimensions from VLAs, and the score distribution in Sim2Real, First/Third perspectives, Main object, and Background sub-categories.

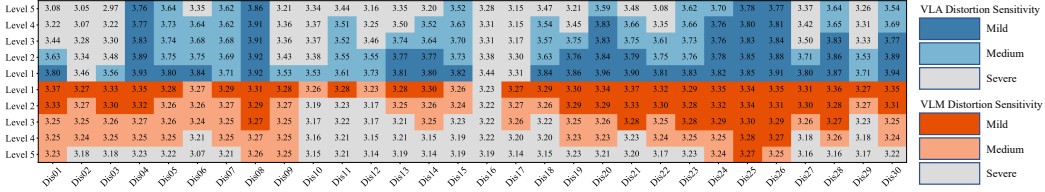

Figure 7: Just-Noticable-Difference (JND) of **VLMs** and **VLAs**. Distortion order follows Figure 5.

## 5 EXPERIMENT

### 5.1 EXPERIMENT SETUPS

We randomly partitioned the Embodied-IQA database into the train/val set, with 29,520 and 7,380 reference/distorted image pairs according to an 8:2 ratio. Since VLA is more downstream than VLM in the Mertonian system of Embodied AI, and previous machine-oriented IQA works are all based on VLM, we use the Decision scores from VLA in the main experiment and put the Cognition scores from VLM in the supplementary. 15 objective IQA metrics are implemented to predict the Decision score, including: (1) 5 baseline metrics: PSNR, SSIM Wang (2004), Brisque Mittal et al. (2012), Q-Align Wu et al. (2024), and Q-Align+ Zhang et al. (2025). Q-Align and Q-Align+ are loaded in quality/aesthetic weights. As the most commonly used IQA metrics, they are performed in a Zero-shot setting; (2) 5 Full-Reference (FR) metrics: AHIQ Lao et al. (2022), CKDN Zheng et al. (2021), DISTS Ding et al. (2020), LPIPS Zhang et al. (2018a), and TOPIQ-FR Chen et al. (2024a). Those learning-based metrics are fine-tuned on our training set, which takes both reference/distorted images as inputs; (3) 5 No-Reference (NR) metrics: CLIPIQA Wang et al. (2023), CNNIQA Kang et al. (2014), DBCNN Zhang et al. (2018b), QualiClip Agnolucci et al. (2025), and TOPIQ-NR Chen et al. (2024a). They are also fine-tuned on a training set, which takes only distorted images as inputs.

To benchmark the performance of quality metrics, three global indicators were employed: SRCC, Kendall Rank-order Correlation Coefficient (KRCC), and Pearson Linear Correlation Coefficient (PLCC), to evaluate the consistency between the objective quality score and the subjective MOS. Among these, SRCC and KRCC represent the prediction monotonicity, while PLCC measures the accuracy. We train the FR/NR metrics on Embodied-IQA with the learning rate as $10^{-5}$ for 50 epochs, under the default settings in pyiqa toolbox, and evaluate the performance on (1) 3 scoring dimension; (2) 3 JND-based distortion sensitivity in Figure 7; (3) First/Third person perspective; (4) Sim2Real; (5) 5 Distortion level. The partitioning, training, and testing pipeline is repeated 10 times, and the mean value is reported as the experimental result. The perception module is based on the Intel RealSense D455 array, supporting both First-person (wrist) and Third-person (top, side) as input. Cognition and Decision annotations are collected on two servers with 16×NVIDIA A800 SXM4 80GB GPUs, and then conduct IQA training/validation on one GPU above. Execution is achieved through the UR5 robotic arm and Robotiq 2F-140 gripper, with a working radius of 85cm.

### 5.2 RESULT AND DISCUSSION

Table 2 presents the performance of advanced quality metrics on the Embodied-IQA database. For the three dimensions of the 7-DoF VLA output, Position is the easiest to predict, followed by State, with Rotation being the most difficult. The SRCC of FR IQA methods with subjective labels is less than 0.65, while NR is even less than 0.6. Note that these methods have a correlation close to 0.9

Table 2: Using 15 advanced IQA metrics to evaluate the Decision score from VLAs, including zero-shot, FR, and NR metrics. [Keys: Best/Second best in group; **Baseline**; Lower than baseline.]

| Dimension | | Position | | | Rotation | | | State | | | First Perspective | | | Third Perspective | | |
|---|---|---|---|---|---|---|---|---|---|---|---|---|---|---|---|---|
| Group | Metric | SRCC↑ | KRCC↑ | PLCC↑ | SRCC↑ | KRCC↑ | PLCC↑ | SRCC↑ | KRCC↑ | PLCC↑ | SRCC↑ | KRCC↑ | PLCC↑ | SRCC↑ | KRCC↑ | PLCC↑ |
| Zero | PSNR | 0.2762 | 0.1872 | 0.3094 | 0.2594 | 0.1756 | 0.3035 | 0.2284 | 0.1522 | 0.2271 | 0.4059 | 0.2763 | 0.4373 | 0.3949 | 0.2693 | 0.4376 |
| | SSIM Wang (2004) | 0.4862 | 0.3345 | 0.4438 | 0.4246 | 0.2891 | 0.3912 | 0.3607 | 0.2468 | 0.3216 | 0.5834 | 0.4101 | 0.5256 | 0.5478 | 0.3815 | 0.5132 |
| | Brisque Mittal et al. (2012) | 0.3073 | 0.2051 | 0.2707 | 0.2752 | 0.1826 | 0.2519 | 0.3335 | 0.2255 | 0.2986 | 0.3302 | 0.2210 | 0.2634 | 0.4020 | 0.2688 | 0.3836 |
| | Q-Align Wu et al. (2024) | **0.5325** | **0.3641** | **0.4869** | **0.5387** | **0.3758** | **0.5329** | **0.3791** | **0.2552** | **0.3346** | **0.6658** | **0.4715** | **0.5992** | **0.5854** | **0.4030** | **0.5578** |
| | Q-Align+ Zhang et al. (2025) | 0.3275 | 0.2157 | 0.2410 | 0.3596 | 0.2465 | 0.2818 | 0.1492 | 0.0972 | 0.1272 | 0.4663 | 0.3167 | 0.4283 | 0.3104 | 0.2021 | 0.2649 |
| FR | AHIQ Lao et al. (2022) | 0.7481 | 0.5496 | 0.7467 | 0.6454 | 0.4655 | 0.6435 | 0.6465 | 0.4609 | 0.6590 | 0.8011 | 0.6014 | 0.8025 | 0.7989 | 0.6007 | 0.7959 |
| | CKDN Zheng et al. (2021) | 0.6748 | 0.4807 | 0.6771 | 0.6061 | 0.4278 | 0.6001 | 0.6324 | 0.4515 | 0.6410 | 0.7716 | 0.5720 | 0.7610 | 0.7641 | 0.5624 | 0.7596 |
| | DISTS Ding et al. (2020) | 0.5797 | 0.4010 | 0.5846 | 0.5366 | 0.3746 | 0.5390 | 0.4653 | 0.3180 | 0.4611 | 0.6458 | 0.4624 | 0.6249 | 0.6545 | 0.4654 | 0.6642 |
| | LPIPS Zhang et al. (2018a) | 0.3922 | 0.2642 | 0.3511 | 0.2972 | 0.1994 | 0.2378 | 0.4210 | 0.2890 | 0.3852 | 0.4697 | 0.3205 | 0.4168 | 0.4821 | 0.3313 | 0.4805 |
| | TOPIQ-FR Chen et al. (2024a) | 0.7748 | 0.5794 | 0.7827 | 0.6428 | 0.4607 | 0.6480 | 0.6684 | 0.4826 | 0.6727 | 0.8307 | 0.6371 | 0.8297 | 0.8322 | 0.6404 | 0.8298 |
| NR | CLIPIQA Wang et al. (2023) | 0.1784 | 0.1172 | 0.1246 | 0.0708 | 0.0193 | 0.0468 | 0.1348 | 0.0770 | 0.0622 | 0.0048 | 0.0043 | 0.0821 | 0.2155 | 0.1287 | 0.1415 |
| | CNNIQA Kang et al. (2014) | 0.5189 | 0.3642 | 0.5318 | 0.4618 | 0.3221 | 0.4587 | 0.4601 | 0.3203 | 0.4667 | 0.5441 | 0.3770 | 0.5618 | 0.6407 | 0.4579 | 0.6468 |
| | DBCNN Zhang et al. (2018b) | 0.6045 | 0.4303 | 0.6094 | 0.5325 | 0.3687 | 0.5341 | 0.5419 | 0.3761 | 0.5441 | 0.6399 | 0.4593 | 0.6408 | 0.6565 | 0.4653 | 0.6609 |
| | QualiClip Agnolucci et al. (2025) | 0.6463 | 0.4619 | 0.6643 | 0.5387 | 0.3768 | 0.5384 | 0.5589 | 0.3912 | 0.5518 | 0.5428 | 0.3870 | 0.5490 | 0.7208 | 0.5240 | 0.7175 |
| | TOPIQ-NR Chen et al. (2024a) | 0.7496 | 0.5549 | 0.7606 | 0.5981 | 0.4253 | 0.6020 | 0.7036 | 0.5100 | 0.6960 | 0.7791 | 0.5804 | 0.7810 | 0.8269 | 0.6341 | 0.8211 |

| Dimension | | Mild Distortion | | | Medium Distortion | | | Severe Distortion | | | Real-world | | | Simulation | | |
|---|---|---|---|---|---|---|---|---|---|---|---|---|---|---|---|---|
| Group | Metric | SRCC↑ | KRCC↑ | PLCC↑ | SRCC↑ | KRCC↑ | PLCC↑ | SRCC↑ | KRCC↑ | PLCC↑ | SRCC↑ | KRCC↑ | PLCC↑ | SRCC↑ | KRCC↑ | PLCC↑ |
| Zero | PSNR | 0.3518 | 0.2396 | **0.4935** | 0.2292 | 0.1555 | 0.2175 | 0.1190 | 0.0806 | 0.1443 | 0.3794 | 0.2575 | 0.3767 | 0.2617 | 0.1778 | 0.3905 |
| | SSIM Wang (2004) | **0.3778** | **0.2620** | 0.2581 | 0.1993 | 0.1333 | 0.1617 | 0.2387 | 0.1604 | 0.2754 | 0.5405 | 0.3754 | 0.4848 | 0.5336 | 0.3724 | 0.4990 |
| | Brisque Mittal et al. (2012) | 0.2088 | 0.1334 | 0.1502 | 0.1503 | 0.0967 | 0.1168 | 0.1525 | 0.1017 | 0.1205 | 0.4143 | 0.2798 | 0.3907 | 0.0636 | 0.0421 | 0.0423 |
| | Q-Align Wu et al. (2024) | 0.3541 | 0.2403 | 0.3097 | **0.4090** | **0.2873** | **0.3899** | **0.3258** | **0.2206** | **0.3723** | **0.6179** | **0.4302** | **0.5639** | **0.6565** | **0.4690** | **0.6333** |
| | Q-Align+ Zhang et al. (2025) | 0.1699 | 0.1132 | 0.1375 | 0.2799 | 0.1908 | 0.2287 | 0.0618 | 0.0425 | 0.1506 | 0.4167 | 0.2753 | 0.3820 | 0.4846 | 0.3285 | 0.4711 |
| FR | AHIQ Lao et al. (2022) | 0.6683 | 0.4831 | 0.6932 | 0.6653 | 0.4821 | 0.7130 | 0.7218 | 0.5308 | 0.7278 | 0.8138 | 0.6223 | 0.8270 | 0.7515 | 0.5583 | 0.7520 |
| | CKDN Zheng et al. (2021) | 0.5733 | 0.4060 | 0.5743 | 0.6610 | 0.4841 | 0.6979 | 0.6909 | 0.5024 | 0.6867 | 0.7227 | 0.5303 | 0.7311 | 0.7676 | 0.5769 | 0.7769 |
| | DISTS Ding et al. (2020) | 0.4564 | 0.3191 | 0.4507 | 0.2343 | 0.1604 | 0.2735 | 0.3177 | 0.2155 | 0.3731 | 0.6443 | 0.4551 | 0.6394 | 0.6408 | 0.4589 | 0.6560 |
| | LPIPS Zhang et al. (2018a) | 0.3004 | 0.1986 | 0.2102 | 0.4208 | 0.2884 | 0.4581 | 0.4672 | 0.3220 | 0.4719 | 0.3605 | 0.2429 | 0.3999 | 0.4665 | 0.3181 | 0.4442 |
| | TOPIQ-FR Chen et al. (2024a) | 0.7128 | 0.5257 | 0.7626 | 0.7238 | 0.5371 | 0.7581 | 0.7355 | 0.5434 | 0.7434 | 0.8104 | 0.6210 | 0.8250 | 0.7815 | 0.5910 | 0.8053 |
| NR | CLIPIQA Wang et al. (2023) | 0.0848 | 0.0563 | 0.0568 | 0.0413 | 0.0300 | 0.0779 | 0.2198 | 0.1509 | 0.2150 | 0.1576 | 0.1040 | 0.1293 | 0.1298 | 0.0863 | 0.1093 |
| | CNNIQA Kang et al. (2014) | 0.3766 | 0.2607 | 0.3847 | 0.3367 | 0.2314 | 0.3907 | 0.3921 | 0.2696 | 0.3957 | 0.5651 | 0.3967 | 0.5875 | 0.5835 | 0.4117 | 0.5901 |
| | DBCNN Zhang et al. (2018b) | 0.4488 | 0.3056 | 0.4309 | 0.4283 | 0.2994 | 0.4699 | 0.4622 | 0.3195 | 0.4459 | 0.6728 | 0.4833 | 0.6806 | 0.6468 | 0.4613 | 0.6345 |
| | QualiClip Agnolucci et al. (2025) | 0.4941 | 0.3425 | 0.4794 | 0.4390 | 0.3094 | 0.4607 | 0.3752 | 0.2633 | 0.3917 | 0.6712 | 0.4882 | 0.6952 | 0.6308 | 0.4468 | 0.6292 |
| | TOPIQ-NR Chen et al. (2024a) | 0.7035 | 0.5164 | 0.7263 | 0.7174 | 0.5312 | 0.7374 | 0.7227 | 0.5312 | 0.7310 | 0.7995 | 0.5771 | 0.8148 | 0.7697 | 0.5771 | 0.7777 |

| Dimension | | Dis-level-1 | | | Dis-level-2 | | | Dis-level-3 | | | Dis-level-4 | | | Dis-level-5 | | |
|---|---|---|---|---|---|---|---|---|---|---|---|---|---|---|---|---|
| Group | Metric | SRCC↑ | KRCC↑ | PLCC↑ | SRCC↑ | KRCC↑ | PLCC↑ | SRCC↑ | KRCC↑ | PLCC↑ | SRCC↑ | KRCC↑ | PLCC↑ | SRCC↑ | KRCC↑ | PLCC↑ |
| Zero | PSNR | 0.2932 | 0.1987 | 0.3856 | 0.2322 | 0.1532 | 0.2837 | 0.2215 | 0.1484 | 0.2749 | 0.2379 | 0.1591 | 0.2642 | 0.3575 | 0.2444 | 0.3571 |
| | SSIM Wang (2004) | 0.4397 | 0.3035 | 0.4038 | 0.4638 | 0.3198 | 0.4208 | 0.4865 | 0.3329 | 0.4446 | 0.4927 | 0.3383 | 0.4745 | 0.5558 | **0.3832** | **0.5327** |
| | Brisque Mittal et al. (2012) | 0.2650 | 0.1786 | 0.2472 | 0.2446 | 0.1618 | 0.2443 | 0.3516 | 0.2346 | 0.3033 | 0.2928 | 0.1933 | 0.2604 | 0.3632 | 0.2434 | 0.3269 |
| | Q-Align Wu et al. (2024) | **0.5049** | **0.3488** | **0.4924** | **0.5534** | **0.3865** | **0.4951** | **0.5609** | **0.3876** | **0.5132** | **0.5753** | **0.4015** | **0.5124** | **0.5567** | 0.3818 | 0.5275 |
| | Q-Align+ Zhang et al. (2025) | 0.2909 | 0.1923 | 0.2210 | 0.4074 | 0.2763 | 0.3245 | 0.3828 | 0.2537 | 0.3241 | 0.3862 | 0.2585 | 0.3145 | 0.3274 | 0.2200 | 0.2694 |
| FR | AHIQ Lao et al. (2022) | 0.7453 | 0.5531 | 0.7722 | 0.7610 | 0.5671 | 0.7741 | 0.7389 | 0.5397 | 0.7546 | 0.7486 | 0.5518 | 0.7692 | 0.7655 | 0.5698 | 0.7820 |
| | CKDN Zheng et al. (2021) | 0.6253 | 0.4513 | 0.6806 | 0.6612 | 0.4785 | 0.6888 | 0.7036 | 0.5106 | 0.7124 | 0.7030 | 0.5131 | 0.7035 | 0.7194 | 0.5271 | 0.7290 |
| | DISTS Ding et al. (2020) | 0.5659 | 0.3962 | 0.5710 | 0.5774 | 0.4079 | 0.5753 | 0.5611 | 0.3903 | 0.5610 | 0.5431 | 0.3766 | 0.5727 | 0.5834 | 0.4015 | 0.5967 |
| | LPIPS Zhang et al. (2018a) | 0.2736 | 0.1810 | 0.2813 | 0.3723 | 0.2550 | 0.4003 | 0.4354 | 0.2927 | 0.4208 | 0.4619 | 0.3156 | 0.4567 | 0.4824 | 0.3322 | 0.4526 |
| | TOPIQ-FR Chen et al. (2024a) | 0.7861 | 0.5939 | 0.8034 | 0.7765 | 0.5828 | 0.8012 | 0.7741 | 0.5809 | 0.7871 | 0.7629 | 0.5682 | 0.7773 | 0.7763 | 0.5766 | 0.7873 |
| NR | CLIPIQA Wang et al. (2023) | 0.1541 | 0.1054 | 0.0822 | 0.1204 | 0.0732 | 0.0635 | 0.1291 | 0.0831 | 0.0821 | 0.1564 | 0.1041 | 0.1129 | 0.1130 | 0.0732 | 0.0846 |
| | CNNIQA Kang et al. (2014) | 0.4871 | 0.3362 | 0.5207 | 0.5215 | 0.3669 | 0.5522 | 0.5441 | 0.3835 | 0.5710 | 0.5433 | 0.3787 | 0.5570 | 0.6083 | 0.4274 | 0.6078 |
| | DBCNN Zhang et al. (2018b) | 0.5982 | 0.4225 | 0.6213 | 0.6194 | 0.4431 | 0.6210 | 0.6235 | 0.4386 | 0.6146 | 0.6189 | 0.4394 | 0.6163 | 0.6141 | 0.4335 | 0.6282 |
| | QualiClip Agnolucci et al. (2025) | 0.5805 | 0.4137 | 0.5787 | 0.6029 | 0.4342 | 0.6389 | 0.6044 | 0.4372 | 0.6101 | 0.6333 | 0.4509 | 0.6469 | 0.6030 | 0.4265 | 0.6399 |
| | TOPIQ-NR Chen et al. (2024a) | 0.7547 | 0.5626 | 0.7781 | 0.7868 | 0.5937 | 0.7989 | 0.7566 | 0.5601 | 0.7695 | 0.7480 | 0.5532 | 0.7658 | 0.7529 | 0.5558 | 0.7670 |

with HVS in traditional human-oriented IQA tasks, which is sufficiently excellent, but they cannot adapt to the database we proposed, indicating that IQA for Embodied AI needs further research. For distortion sensitivity, the Decision scores are high under mild distortions, with small internal gaps and difficulty in prediction; whereas when the distortion becomes severe, the Distortion scores fluctuate more, resulting in a higher SRCC. For Perspective and Sim2Real, most IQA methods perform better on third-person, real images. Therefore, in Embodied scenarios, more content captured by the robotic arm itself or from simulation software should be used. For the five absolute distortion levels, the performance of IQA methods does not change much. This further proves that dividing distortion levels based on HVS is insufficient, and the distortion levels of Embodied AI should be divided by the JND of RVS itself, as we have done in the Embodied-IQA database. Comparing various IQA methods longitudinally, FR is superior to NR in most cases, with TOPIQ maintaining the leading performance in most cases, with an SRCC of about 0.75, which still needs improvement compared to human-oriented IQA. It is worth mentioning that the main parameters of some methods have been frozen based on HVS, such as LPIPS, DISTS, and CLIPIQA. Thus, although after training, they are even worse than the zero-shot baseline. This further reflects the gap between HVS and RVS, implying the significance of proposing the Embodied IQA task.

## 5.3 CROSS DATABASE VALIDATION

To further analyze the performance of RVS-oriented IQA on HVS/MVS, we conducted cross-validation using Embodied-IQA VLA Decision score for training, VLM Cognition score, and two of

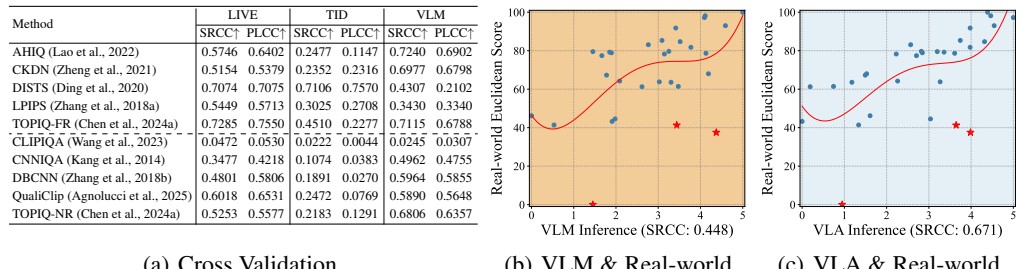

| Method | LIVE | | TID | | VLM | |
|---|---|---|---|---|---|---|
| | SRCC↑ | PLCC↑ | SRCC↑ | PLCC↑ | SRCC↑ | PLCC↑ |
| AHIQ (Lao et al., 2022) | 0.5746 | 0.6402 | 0.2477 | 0.1147 | 0.7240 | 0.6902 |
| CKDN (Zheng et al., 2021) | 0.5154 | 0.5379 | 0.2352 | 0.2316 | 0.6977 | 0.6798 |
| DISTS (Ding et al., 2020) | 0.7074 | 0.7075 | 0.7106 | 0.7570 | 0.4307 | 0.2102 |
| LPIPS (Zhang et al., 2018a) | 0.5449 | 0.5713 | 0.3025 | 0.2708 | 0.3430 | 0.3340 |
| TOPIQ-FR (Chen et al., 2024a) | 0.7285 | 0.7550 | 0.4510 | 0.2277 | 0.7115 | 0.6788 |
| CLIPIQA (Wang et al., 2023) | 0.0472 | 0.0530 | 0.0222 | 0.0044 | 0.0245 | 0.0307 |
| CNNIQA (Kang et al., 2014) | 0.3477 | 0.4218 | 0.1074 | 0.0383 | 0.4962 | 0.4755 |
| DBCNN (Zhang et al., 2018b) | 0.4801 | 0.5806 | 0.1891 | 0.0270 | 0.5964 | 0.5855 |
| QualiClip (Agnolucci et al., 2025) | 0.6018 | 0.6531 | 0.2472 | 0.0769 | 0.5890 | 0.5648 |
| TOPIQ-NR (Chen et al., 2024a) | 0.5253 | 0.5577 | 0.2183 | 0.1291 | 0.6806 | 0.6357 |

(a) Cross Validation  (b) VLM & Real-world  (c) VLA & Real-world

Figure 8: Cross database validation on Cognition and human-oriented score, and the correlation between VLM&VLA and Real-world. ★ denotes distortions with greater Real-world impact.

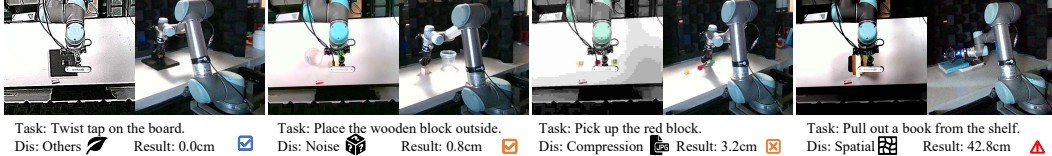

Task: Twist tap on the board. Dis: Others Result: 0.0cm    Task: Place the wooden block outside. Dis: Noise Result: 0.8cm    Task: Pick up the red block. Dis: Compression Result: 3.2cm    Task: Pull out a book from the shelf. Dis: Spatial Result: 42.8cm

Figure 9: Positive/Negative cases of real-world experiment. The score of successful examples is 100, and deduct the Euclidean distance for failed examples. Triggering interruption will be scored as 0.

the most commonly human-oriented databases, LIVE2008 Moorthy & Bovik (2011) and TID2013 Ponomarenko et al. (2015) for validation, employing the same parameter settings as Section 5.1. According to the results in Table 8 (a), IQA methods fine-tuned on Embodied AI data lose certain human-oriented evaluation capabilities, where the SRCC is even lower than 0.4 in the LIVE database. Fortunately, the IQA model trained with VLA annotations can also predict VLM scores, and the SRCC of AHIQ can reach 0.7, revealing the internal connection between Cognition and Decision.

## 5.4 REAL-WORLD EXPERIMENT

Since Embodied AI is ultimately applied in the Real-world, we compare Execution with Cognition/Decision to link External and Internal Reality, thereby proving the reliability of the 5m+ annotations in the Embodied-IQA database. Specifically, we selected 5 VLA that support multi-step output and executed the 10 tasks in Section 3.2 on 30 types of distorted images. Note that among the five difficulty levels in Perception, we only executed the simplest one to ensure that the execution result of the reference image is correct. Thus, we ensured that the reason for execution failure came from the added distortion, not the image itself. Figure 9 shows examples of successful execution, results deviating from the ground truth, and emergency stops triggered by collisions with the table. We calculate the average Execution score under 30 distortion types and compare it with the Cognition and Decision scores, as shown in Figure 8 (b)(c), where findings are summarized as follows:

Cognition VS Execution: The SRCC of VLM results with the real world is less than 0.5. This corroborates the necessity of using VLA as subjects in the Embodied IQA task beyond VLM.
Decision VS Execution: The SRCC of VLA results with the real world exceeds 0.6, indicating that Decision can represent Execution to some extent. However, this correlation is still not high enough, proving that certain real-world experiments are still indispensable for Embodied AI development.
Perception VS Cognition&Decision: Existing quality metrics have initially demonstrated the ability to handle Embodied IQA tasks, but there is still a gap compared to the traditional human-oriented paradigm. More advanced metrics should be developed in the upcoming Embodied AI era.

## 6 CONCLUSION

In this paper, we extend the application of IQA from a traditional human-oriented paradigm to Embodied AI. To study which distortions have a negative impact on Embodied AI, we built a Perception-Cognition-Decision-Execution pipeline based on Mertonian Law and established a database for Embodied subjective preferences. Experiments show that more advanced IQA methods are needed to identify quality degradation for Embodied AI. We sincerely hope this Embodied IQA task can promote the application of Robotic Intelligence under complex distortions in the Real-world.

## ACKNOWLEDGMENT

This work was supported in part by the National Natural Science Foundation of China under Grants 625B2118, 62501387, 62572317, 62501337, 62225112, in part by the Singapore Ministry of Education under Grant ZDSYS20220527171406015, and in part by the New Generation Artificial Intelligence-National Science and Technology Major Project (2025ZD0124104) in collaboration with Shanghai Artificial Intelligence Laboratory.

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

## A    STATEMENT

**[LLM Usage]**: LLM is applied to aid and polish writing, including grammar, rhetoric, and proper citation format. No scientific conclusions, experimental designs, or technical contributions are generated by LLM.

**[Ethics statement]**: This article covers Embodied AI's four steps: perception, cognition, decision, and execution. Most experiments focus on machine intelligence and humans are involved only in the perception annotation phases. We obtained consent from each participant before data collection and ensured that the experimental procedures adhered to the Declaration of Helsinki. For Embodied AI execution, we implemented an emergency stop mechanism for the robotic arm to ensure compliance with the Three AI Laws (do no harm to humans/the environment/itself).

**[Reproducibility]**: We will release all quality-related issue for non-commercial use, including the quality score from each VLMs/VLAs, the fidelity of position/rotation/state, and the overall score.

## B    LIMITATION & BROADER IMPACT

**[Limitation 1]**: As the first Embodied IQA work, we simplify the Perception to vision and Execution to robotic arms. Since the visual signals processed by humans account for 80% of the total signals, while upper limb movements account for 50% of all movements. Considering the consistency between humans and machines and the current limitations of Embodied AI, our simplification is reasonable. This will not affect the current main applications of Embodied AI, such as industrial assembly and home services. After the future vision-tactile fusion (Perception), quadruped robot dog (Execution), and other task scenarios are improved, we will further update the quality assessment data.

**[Limitation 2]**: The scale of our Real-world data is relatively small compared to Cognition and Decision. In the VLM and VLA steps, we already have the largest amount of data (millions) compared to previous IQA work. This is because the high cost of real machine data requires a lot of manpower in the site layout and verification stages. Although 1,500 annotation samples are not enough as quality labels, they are sufficient to verify the Sim2Real consistency of Cognition-Execution and Decision-Execution. With the further development of Embodied AI, we believe an automated Real-world pipeline will be developed, from which we will expand the scale of Execution labels.

**[Broader Impact] (Positive)**: IQA can expand the application scenarios of Embodied AI, extending it from the in-lab environment to distortions in the Real-world. We collect the subjective preferences of Embodied AI, thus objectively judge the 'utility' of images before executing specific tasks. In this way, distorted images such as jitter and blur can be effectively filtered. Such quality indicators can be used for all visual applications for Embodied AI, such as video coding, super-resolution, defogging, denoising, etc. Considering that the amount of visual signals consumed by machines has exceeded humans since 2023, visual quality indicators for Embodied AI can fill this research gap.

**[Broader Impact] (Negative)**: The general use of visual quality indicators in Embodied AI may affect traditional human-oriented tasks. Considering that humans, VLM (Cognition), and VLA (Decision) have different preferences, only evaluating the preferences of VLM and VLA will inevitably lead to scores that are not relevant to humans. Therefore, in future international protocols, it is recommended to integrate the three IQA paradigms for humans, VLM, and VLA together, and select appropriate quality indicators based on the user end.

## C    ROBOTICS SETTINGS

This section provides a detailed derivation of the forward and inverse kinematics for the Universal Robots (UR5), a 6-DoF collaborative robot. The Denavit-Hartenberg (D-H) convention is used to establish the kinematic model.

In the experiments part, the initial pose is obtained through forward kinematics by recording the initial rotational angles of the six joints and calculating the end-effector's pose relative to the base coordinate frame. The incremental pose output by the VLA is then multiplied with the initial pose to derive the step-by-step poses. The robotic arm's actual motion is resolved via inverse kinematics, which computes the required rotational angles for each joint motor to achieve the target configuration.

Table 3: Parameter settings of Robotic arm UR5 D-H. The specific action depends on 6 frames.

| Joint Frame $i$ | $\alpha_{i-1}$ (rad) | $a_{i-1}$ (m) | $d_i$ (m) | $\theta_i$ (rad) |
|---|---|---|---|---|
| 1 | 0 | 0 | $d_1$ | $\theta_1^*$ |
| 2 | $\pi/2$ | 0 | 0 | $\theta_2^*$ |
| 3 | 0 | $a_2$ | 0 | $\theta_3^*$ |
| 4 | 0 | $a_3$ | $d_4$ | $\theta_4^*$ |
| 5 | $\pi/2$ | 0 | $d_5$ | $\theta_5^*$ |
| 6 | $-\pi/2$ | 0 | $d_6$ | $\theta_6^*$ |

The D-H parameters define the geometry of the robot manipulator by establishing a coordinate frame $\{i\}$ attached to each link $i$. The transformation from frame $\{i-1\}$ to frame $\{i\}$, denoted $A_i^{i-1}$, is described by four parameters associated with link $i-1$ and joint $i$:

- $\theta_i$: Joint Angle - the rotation about the $z_{i-1}$ axis, from $x_{i-1}$ to $x_i$. For a revolute joint, $\theta_i$ is the joint variable.

- $d_i$: Link Offset - the distance along the $z_{i-1}$ axis from the origin of frame $\{i-1\}$ to the intersection of the $z_{i-1}$ axis with the $x_i$ axis. For a prismatic joint, $d_i$ is the joint variable.

- $a_{i-1}$: Link Length - the distance along the $x_i$ axis (which is the common normal between $z_{i-1}$ and $z_i$) from the intersection of $z_{i-1}$ and $x_i$ axis to the origin of frame $\{i\}$.

- $\alpha_{i-1}$: Link Twist - the angle about the $x_i$ axis, from $z_{i-1}$ to $z_i$.

The UR5 D-H parameters used in this paper shown in Table 3. Where $a_2, a_3$ are physical link lengths associated with links 2 and 3 respectively (used as $a_{i-1}$ parameters in the table for joints 3 and 4), and $d_1, d_4, d_5, d_6$ are link offsets. The $\theta_i^*$ are the joint variables.

Typical UR5 parameter values (example, signs depend on coordinate frame choices): $d_1 = 0.089159$ m, $a_2 = 0.42500$ m (often negative in some tables: $-0.42500$), $a_3 = 0.39225$ m (often negative: $-0.39225$), $d_4 = 0.10915$ m, $d_5 = 0.09465$ m, $d_6 = 0.0823$ m.

## D  FORWARD KINEMATICS

The standard D-H transformation matrix $A_i^{i-1}$ from frame $\{i-1\}$ to frame $\{i\}$ is defined as a product of four basic transformations:

$$A_i^{i-1} = \mathbf{R}_z(\theta_i)\mathbf{Tr}_z(d_i)\mathbf{Tr}_x(a_{i-1})\mathbf{R}_x(\alpha_{i-1}), \tag{1}$$

$$A_i^{i-1} = \begin{pmatrix} \cos(\theta_i) & -\sin(\theta_i)\cos(\alpha_{i-1}) & \sin(\theta_i)\sin(\alpha_{i-1}) & a_{i-1}\cos(\theta_i) \\ \sin(\theta_i) & \cos(\theta_i)\cos(\alpha_{i-1}) & -\cos(\theta_i)\sin(\alpha_{i-1}) & a_{i-1}\sin(\theta_i) \\ 0 & \sin(\alpha_{i-1}) & \cos(\alpha_{i-1}) & d_i \\ 0 & 0 & 0 & 1 \end{pmatrix}, \tag{2}$$

where $\mathbf{Tr}(\cdot)$ and $\mathbf{R}(\cdot)$ denote the trajectory and rotation matrix projected on a certain axis. For simplicity, the following symbols will be defined in the subsequent sections: $c_i = \cos(\theta_i)$, $s_i = \sin(\theta_i)$. $c_{ij} = \cos(\theta_i + \theta_j)$, $s_{ij} = \sin(\theta_i + \theta_j)$. Using the D-H parameters from Table 3, the individual transformation matrix $A_i^{i-1}$ for robotic manipulators are:

$$A_1^0 = \begin{pmatrix} \cos(\theta_1) & -\sin(\theta_1) & 0 & 0 \\ \sin(\theta_1) & \cos(\theta_1) & 0 & 0 \\ 0 & 0 & 1 & d_1 \\ 0 & 0 & 0 & 1 \end{pmatrix}, \tag{3}$$

$$A_2^1 = \begin{pmatrix} \cos(\theta_2) & 0 & \sin(\theta_2) & 0 \\ \sin(\theta_2) & 0 & -\cos(\theta_2) & 0 \\ 0 & 1 & 0 & 0 \\ 0 & 0 & 0 & 1 \end{pmatrix}, \tag{4}$$

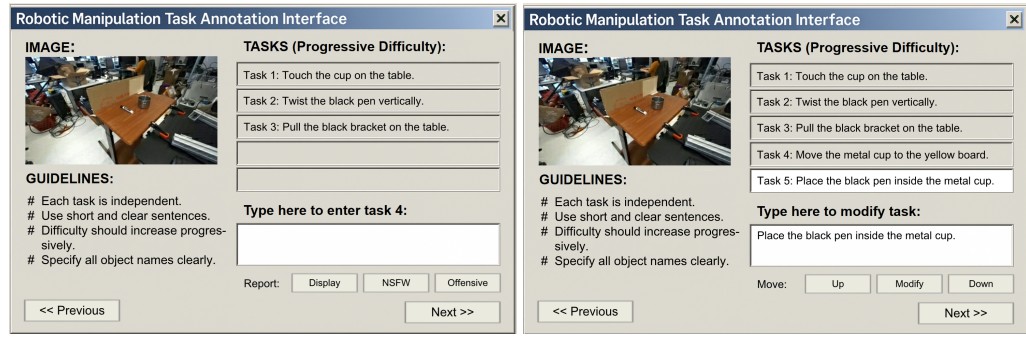

(a) Annotation $\qquad$ (b) Verification

Figure 10: Human annotation and verification Interface. Subjects will cyclically define five tasks with different contents and increasing difficulty, and submit them to robotic experts for verification.

$$A_3^2 = \begin{pmatrix} \cos(\theta_3) & -\sin(\theta_3) & 0 & a_2\cos(\theta_3) \\ \sin(\theta_3) & \cos(\theta_3) & 0 & a_2\sin(\theta_3) \\ 0 & 0 & 1 & 0 \\ 0 & 0 & 0 & 1 \end{pmatrix}, \tag{5}$$

$$A_4^3 = \begin{pmatrix} \cos(\theta_4) & -\sin(\theta_4) & 0 & a_3\cos(\theta_4) \\ \sin(\theta_4) & \cos(\theta_4) & 0 & a_3\sin(\theta_4) \\ 0 & 0 & 1 & d_4 \\ 0 & 0 & 0 & 1 \end{pmatrix}, \tag{6}$$

$$A_5^4 = \begin{pmatrix} \cos(\theta_5) & 0 & \sin(\theta_5) & 0 \\ \sin(\theta_5) & 0 & -\cos(\theta_5) & 0 \\ 0 & 1 & 0 & d_5 \\ 0 & 0 & 0 & 1 \end{pmatrix}, \tag{7}$$

$$A_6^5 = \begin{pmatrix} \cos(\theta_6) & 0 & -\sin(\theta_6) & 0 \\ \sin(\theta_6) & 0 & \cos(\theta_6) & 0 \\ 0 & -1 & 0 & d_6 \\ 0 & 0 & 0 & 1 \end{pmatrix}. \tag{8}$$

The total transformation matrix $T_0^6$ from the base frame $\{0\}$ to the end-effector frame $\{6\}$ is:

$$T_0^6 = A_1^0 A_2^1 A_3^2 A_4^3 A_5^4 A_6^5 = \begin{pmatrix} R_0^6 & p_0^6 \\ \mathbf{0}_{1\times3} & 1 \end{pmatrix} = \begin{pmatrix} n_x & s_x & a_x & p_x \\ n_y & s_y & a_y & p_y \\ n_z & s_z & a_z & p_z \\ 0 & 0 & 0 & 1 \end{pmatrix}, \tag{9}$$

where $R_0^6 = [n, s, a]$ refers to the rotation matrix part of $T_0^6$, where $n = [n_x, n_y, n_z]^\mathsf{T}$, $s = [s_x, s_y, s_z]^\mathsf{T}$, and $a = [a_x, a_y, a_z]^\mathsf{T}$ are column vectors representing the $x, y, z$ axes of frame $\{6\}$ expressed in frame $\{0\}$, respectively. $p_0^6 = [p_x, p_y, p_z]^\mathsf{T}$ represents the translation vector part of $T_0^6$, namely the position of the origin of frame $\{6\}$ expressed in frame $\{0\}$.

## E  INVERSE KINEMATICS

The objective of Inverse Kinematics (IK) is to determine the set of joint angles $(\theta_1, \ldots, \theta_6)$ that achieve a desired end-effector pose $T_0^6$. The UR5 possesses a spherical wrist (axes of joints 4, 5, and 6 intersect at a common point, the wrist center), which allows for a decoupled analytical solution. First, the position of the wrist center is determined, which allows solving for the first three joints. Then, the orientation of the end-effector is used to solve for the remaining three wrist joints.

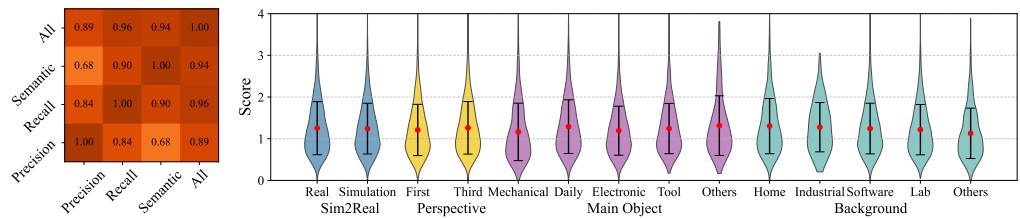

Figure 11: Correlation between the general Cognition score and the 3 dimensions from VLMs, and the score distribution in Sim2Real, First/Third perspectives, Main object, and Background sub-categories.

### E.1 CALCULATION OF THE WRIST CENTER POINT

The wrist center point (WCP), $p_{wc}$, is typically defined as the origin of frame $\{5\}$. Its position can be found by translating from the end-effector origin, $p_0^6$, backwards along the approach vector $a$ (the $z_6$-axis expressed in frame $\{0\}$) by a distance $d_6$:

$$p_{wc} = p_0^6 - R_0^6 \begin{pmatrix} 0 \\ 0 \\ d_6 \end{pmatrix} = \begin{pmatrix} p_x \\ p_y \\ p_z \end{pmatrix} - d_6 \begin{pmatrix} a_x \\ a_y \\ a_z \end{pmatrix} = \begin{pmatrix} p_x - d_6 a_x \\ p_y - d_6 a_y \\ p_z - d_6 a_z \end{pmatrix}, \tag{10}$$

where $p_{wc} = (x_{wc}, y_{wc}, z_{wc})^\mathsf{T}$ means position vector of the wrist center point in frame $\{0\}$.

### E.2 SOLVING FOR BASE JOINTS

The $y$-coordinate of $p_{wc}$ when expressed in frame $\{1\}$, denoted $p_{wc,y}^1$, can be shown to be $d_4 + d_5$ for this specific D-H parameterization (where $p_{wc}$ is the origin of frame $\{5\}$). We have $p_{wc,y}^1 = -x_{wc}\sin(\theta_1) + y_{wc}\cos(\theta_1) = d_4 + d_5$. This equation can be solved for $\theta_1$:

$$\theta_1 = \operatorname{atan2}(y_{wc}, x_{wc}) - \operatorname{atan2}(d_4 + d_5, \sigma_1 \sqrt{x_{wc}^2 + y_{wc}^2 - (d_4 + d_5)^2}), \tag{11}$$

where $\sigma_1 = \pm 1$ denotes two possible solutions for $\theta_1$. Function $\operatorname{atan2}(\cdot, \cdot)$ transforms two Cartesian coordinates to polar. If the term under the square root is negative, the target $p_{wc}$ is unreachable.

### E.3 SOLVING FOR ELBOW JOINTS

With $\theta_1$ known, transform $p_{wc}$ into frame $\{1\}$. Let $K_x$ and $K_z$ be coordinates of $p_{wc}$ relevant for the planar geometry of links 2 and 3: $K_x = x_{wc}\cos(\theta_1) + y_{wc}\sin(\theta_1)$ $K_z = z_{wc} - d_1$. From the geometry of the first three links (considering $\alpha_1 = \pi/2$ which introduces a rotation making $z_1$ horizontal in the new projected plane if $\theta_2 = 0$): $K_x = a_2\cos(\theta_2) + a_3\cos(\theta_2 + \theta_3)$ $K_z = -a_2\sin(\theta_2) - a_3\sin(\theta_2 + \theta_3)$. Squaring and adding these two equations yields: $K_x^2 + K_z^2 = a_2^2 + a_3^2 + 2a_2a_3\cos(\theta_3)$. This allows solving for $\theta_3$:

$$\cos(\theta_3) = \frac{K_x^2 + K_z^2 - a_2^2 - a_3^2}{2a_2a_3}, \tag{12}$$

where $\sigma_3 = \pm 1$ corresponds to up/down configurations. From $\sin(\theta_3) = \sigma_3\sqrt{1 - \cos(\theta_3)^2}$ we have:

$$\theta_3 = \operatorname{atan2}(\sin(\theta_3), \cos(\theta_3)). \tag{13}$$

To solve for $\theta_2$, rearrange the equations for $K_x$ and $K_z$: Let $k_1 = a_2 + a_3\cos(\theta_3)$ and $k_2 = a_3\sin(\theta_3)$. Then $K_x = k_1\cos(\theta_2) - k_2\sin(\theta_2)$ and $K_z = -k_1\sin(\theta_2) - k_2\cos(\theta_2)$. Solving this system for $\sin(\theta_2)$ and $\cos(\theta_2)$: $\sin(\theta_2) = -(k_1 K_z + k_2 K_x)/(k_1^2 + k_2^2)$ $\cos(\theta_2) = (k_1 K_x - k_2 K_z)/(k_1^2 + k_2^2)$ (Note: $k_1^2 + k_2^2 = K_x^2 + K_z^2$):

$$\theta_2 = \operatorname{atan2}(-(k_1 K_z + k_2 K_x), k_1 K_x - k_2 K_z). \tag{14}$$

Alternatively, a more robust form is often $\theta_2 = \operatorname{atan2}(-K_z, K_x) - \operatorname{atan2}(k_2, k_1)$.

### E.4 SOLVING FOR WRIST JOINTS

Once $\theta_1, \theta_2, \theta_3$ are known, the rotation matrix $R_0^3$ from the base to frame $\{3\}$ can be computed: $R_0^3 = (A_1^0 A_2^1 A_3^2)_{rot}$. The rotation matrix from frame $\{3\}$ to frame $\{6\}$ is then $R_3^6 = (R_0^3)^\mathsf{T} R_0^6$. Let

Table 4: Using 15 advanced IQA metrics to evaluate the **Cognition** score from VLMs, including zero-shot, FR, and NR metrics. [Keys: **Best**/**Second best** in group; **Baseline**; Lower than baseline.]

| Group | Dimension → Metric | Precision SRCC↑ | KRCC↑ | PLCC↑ | Recall SRCC↑ | KRCC↑ | PLCC↑ | Semantic SRCC↑ | KRCC↑ | PLCC↑ | First Perspective SRCC↑ | KRCC↑ | PLCC↑ | Third Perspective SRCC↑ | KRCC↑ | PLCC↑ |
|---|---|---|---|---|---|---|---|---|---|---|---|---|---|---|---|---|
| Zero | PSNR | 0.3432 | 0.2339 | 0.3661 | 0.3186 | 0.2168 | 0.3369 | 0.3257 | 0.2225 | 0.3520 | 0.4142 | 0.2831 | 0.4393 | 0.3605 | 0.2477 | 0.4140 |
| | SSIM Wang (2004) | 0.5809 | 0.4055 | 0.5561 | 0.5558 | 0.3871 | 0.5241 | 0.5798 | 0.4057 | 0.5521 | 0.6244 | 0.4383 | 0.5805 | 0.5849 | 0.4094 | 0.5438 |
| | Brisque Mittal et al. (2012) | 0.3527 | 0.2380 | 0.3342 | 0.3537 | 0.2412 | 0.3319 | 0.3596 | 0.2442 | 0.3375 | 0.3232 | 0.2187 | 0.2987 | 0.3751 | 0.2545 | 0.3554 |
| | Q-Align Wu et al. (2024) | **0.7045** | **0.5067** | **0.6721** | **0.6755** | **0.4798** | **0.6278** | **0.7040** | **0.5058** | **0.6687** | **0.6622** | **0.4767** | **0.6214** | **0.7321** | **0.5349** | **0.6970** |
| | Q-Align+ Zhang et al. (2025) | 0.4697 | 0.3184 | 0.4049 | 0.4524 | 0.3063 | 0.3623 | 0.4722 | 0.3202 | 0.3958 | 0.4687 | 0.3188 | 0.4263 | 0.5082 | 0.3421 | 0.4351 |
| FR | AHIQ Lao et al. (2022) | 0.7941 | 0.5930 | 0.7901 | 0.7747 | 0.5753 | 0.7683 | 0.7983 | 0.5976 | 0.7919 | 0.8035 | 0.6039 | 0.8015 | 0.8288 | 0.6342 | 0.8292 |
| | CKDN Zheng et al. (2021) | 0.7461 | 0.5460 | 0.7444 | 0.7387 | 0.5380 | 0.7332 | 0.7516 | 0.5508 | 0.7470 | 0.7556 | 0.5587 | 0.7547 | 0.7836 | 0.5808 | 0.7810 |
| | DISTS Ding et al. (2020) | 0.7017 | 0.5128 | 0.7052 | 0.6887 | 0.5010 | 0.6863 | 0.7080 | 0.5188 | 0.7096 | 0.7307 | 0.5424 | 0.7299 | 0.7535 | 0.5584 | 0.7530 |
| | LPIPS Zhang et al. (2018a) | 0.6681 | 0.4785 | 0.6165 | 0.6463 | 0.4610 | 0.5975 | 0.6714 | 0.4812 | 0.6179 | 0.6797 | 0.4929 | 0.6292 | 0.6893 | 0.5008 | 0.6485 |
| | TOPIQ-FR Chen et al. (2024a) | 0.8209 | 0.6241 | 0.8194 | 0.8160 | 0.6170 | 0.8126 | 0.8326 | 0.6363 | 0.8289 | 0.7997 | 0.5967 | 0.7932 | 0.8521 | 0.6574 | 0.8462 |
| NR | CLIPIQA Wang et al. (2023) | 0.3111 | 0.2101 | 0.3185 | 0.3013 | 0.2038 | 0.3141 | 0.3167 | 0.2156 | 0.3257 | 0.1748 | 0.1198 | 0.1778 | 0.3889 | 0.2620 | 0.3821 |
| | CNNIQA Kang et al. (2014) | 0.4864 | 0.3359 | 0.4818 | 0.4793 | 0.3312 | 0.4761 | 0.4880 | 0.3368 | 0.4861 | 0.4719 | 0.3247 | 0.4735 | 0.5336 | 0.3744 | 0.5431 |
| | DBCNN Zhang et al. (2018b) | 0.5687 | 0.3921 | 0.5553 | 0.5349 | 0.3650 | 0.5183 | 0.5596 | 0.3835 | 0.5453 | 0.5341 | 0.3578 | 0.5346 | 0.6622 | 0.4699 | 0.6489 |
| | QualiClip Agnolucci et al. (2025) | 0.7416 | 0.5425 | 0.7389 | 0.7399 | 0.5399 | 0.7383 | 0.7524 | 0.5514 | 0.7499 | 0.6960 | 0.5022 | 0.6913 | 0.7864 | 0.5832 | 0.7711 |
| | TOPIQ-NR Chen et al. (2024a) | 0.7941 | 0.5933 | 0.7897 | 0.7818 | 0.5812 | 0.7761 | 0.8031 | 0.6039 | 0.7958 | 0.7854 | 0.5846 | 0.7819 | 0.8512 | 0.6577 | 0.8449 |

| Group | Dimension → Metric | Mild Distortion SRCC↑ | KRCC↑ | PLCC↑ | Medium Distortion SRCC↑ | KRCC↑ | PLCC↑ | Severe Distortion SRCC↑ | KRCC↑ | PLCC↑ | Real-world SRCC↑ | KRCC↑ | PLCC↑ | Simulation SRCC↑ | KRCC↑ | PLCC↑ |
|---|---|---|---|---|---|---|---|---|---|---|---|---|---|---|---|---|
| Zero | PSNR | 0.2350 | 0.1588 | 0.3725 | 0.1122 | 0.0751 | 0.1527 | 0.0811 | 0.0566 | 0.0763 | 0.4390 | 0.3056 | 0.4575 | 0.3613 | 0.2466 | 0.3945 |
| | SSIM Wang (2004) | 0.3263 | 0.2229 | 0.3133 | 0.1434 | 0.0972 | 0.1080 | 0.2102 | 0.1375 | 0.2402 | 0.6319 | 0.4507 | 0.5916 | **0.6645** | **0.4688** | **0.6449** |
| | Brisque Mittal et al. (2012) | 0.1525 | 0.1205 | 0.1017 | 0.1503 | 0.1168 | 0.0967 | 0.2088 | 0.1502 | 0.1334 | 0.4143 | 0.3907 | 0.2798 | 0.0636 | 0.0423 | 0.0421 |
| | Q-Align Wu et al. (2024) | **0.5092** | **0.3572** | **0.4793** | **0.4491** | **0.3071** | **0.4360** | **0.4056** | **0.2775** | **0.5047** | **0.7764** | **0.5739** | **0.7475** | 0.6642 | 0.4648 | 0.6319 |
| | Q-Align+ Zhang et al. (2025) | 0.2995 | 0.2006 | 0.2657 | 0.3422 | 0.2268 | 0.2990 | 0.2154 | 0.1440 | 0.2240 | 0.5643 | 0.3898 | 0.5316 | 0.4804 | 0.3302 | 0.4709 |
| FR | AHIQ Lao et al. (2022) | 0.5921 | 0.4191 | 0.5842 | 0.5538 | 0.3895 | 0.5644 | 0.5580 | 0.3968 | 0.6064 | 0.8293 | 0.6339 | 0.8214 | 0.8138 | 0.6117 | 0.8041 |
| | CKDN Zheng et al. (2021) | 0.5497 | 0.3846 | 0.5672 | 0.5539 | 0.3872 | 0.5537 | 0.5087 | 0.3533 | 0.5287 | 0.7480 | 0.5494 | 0.7457 | 0.7706 | 0.5651 | 0.7727 |
| | DISTS Ding et al. (2020) | 0.5376 | 0.3796 | 0.5179 | 0.2957 | 0.2030 | 0.2894 | 0.3258 | 0.2215 | 0.3657 | 0.7606 | 0.5695 | 0.7566 | 0.7675 | 0.5686 | 0.7677 |
| | LPIPS Zhang et al. (2018a) | 0.4635 | 0.3229 | 0.4285 | 0.3876 | 0.2661 | 0.3700 | 0.3168 | 0.2148 | 0.3225 | 0.7128 | 0.5190 | 0.6493 | 0.6693 | 0.4825 | 0.6030 |
| | TOPIQ-FR Chen et al. (2024a) | 0.6755 | 0.4873 | 0.6661 | 0.6152 | 0.4355 | 0.6194 | 0.5798 | 0.4070 | 0.6125 | 0.8392 | 0.6398 | 0.8232 | 0.8449 | 0.6501 | 0.8442 |
| NR | CLIPIQA Wang et al. (2023) | 0.1542 | 0.1031 | 0.1350 | 0.0375 | 0.0262 | 0.0319 | 0.0974 | 0.0645 | 0.1245 | 0.3944 | 0.2624 | 0.3853 | 0.6178 | 0.4305 | 0.5342 |
| | CNNIQA Kang et al. (2014) | 0.2697 | 0.1797 | 0.2238 | 0.1948 | 0.1310 | 0.1801 | 0.2033 | 0.1382 | 0.1871 | 0.5317 | 0.3732 | 0.5419 | 0.5280 | 0.3659 | 0.5380 |
| | DBCNN Zhang et al. (2018b) | 0.3227 | 0.2200 | 0.3453 | 0.2421 | 0.1591 | 0.2323 | 0.2218 | 0.1480 | 0.2486 | 0.6688 | 0.4743 | 0.6579 | 0.6285 | 0.4306 | 0.5990 |
| | QualiClip Agnolucci et al. (2025) | 0.5795 | 0.4074 | 0.5388 | 0.4769 | 0.3258 | 0.4673 | 0.4846 | 0.3371 | 0.5112 | 0.7847 | 0.5849 | 0.7739 | 0.7691 | 0.5667 | 0.7420 |
| | TOPIQ-NR Chen et al. (2024a) | 0.6443 | 0.4550 | 0.6295 | 0.6006 | 0.4248 | 0.5996 | 0.5703 | 0.4014 | 0.6153 | 0.8403 | 0.6454 | 0.8320 | 0.8322 | 0.6307 | 0.8241 |

| Group | Dimension → Metric | Dis-level-1 SRCC↑ | KRCC↑ | PLCC↑ | Dis-level-2 SRCC↑ | KRCC↑ | PLCC↑ | Dis-level-3 SRCC↑ | KRCC↑ | PLCC↑ | Dis-level-4 SRCC↑ | KRCC↑ | PLCC↑ | Dis-level-5 SRCC↑ | KRCC↑ | PLCC↑ |
|---|---|---|---|---|---|---|---|---|---|---|---|---|---|---|---|---|
| Zero | PSNR | 0.2405 | 0.1665 | 0.3445 | 0.2447 | 0.1644 | 0.2930 | 0.2261 | 0.1512 | 0.2695 | 0.2065 | 0.1357 | 0.2323 | 0.4268 | 0.2894 | 0.4473 |
| | SSIM Wang (2004) | 0.5085 | 0.3520 | 0.4753 | 0.4888 | 0.3389 | 0.4547 | 0.5601 | 0.3878 | 0.5209 | 0.5646 | 0.3852 | 0.5450 | 0.6977 | 0.4997 | **0.6846** |
| | Brisque Mittal et al. (2012) | 0.1471 | 0.0971 | 0.1276 | 0.2153 | 0.1466 | 0.2050 | 0.3106 | 0.2093 | 0.2767 | 0.3112 | 0.2058 | 0.2929 | 0.4560 | 0.3109 | 0.4301 |
| | Q-Align Wu et al. (2024) | **0.6748** | **0.4887** | **0.6579** | **0.6826** | **0.4943** | **0.6539** | **0.7060** | **0.5079** | **0.6656** | **0.7316** | **0.5250** | **0.6937** | **0.7055** | **0.5040** | 0.6775 |
| | Q-Align+ Zhang et al. (2025) | 0.4598 | 0.3136 | 0.4055 | 0.5730 | 0.4006 | 0.5013 | 0.5487 | 0.3758 | 0.4988 | 0.5392 | 0.3684 | 0.5110 | 0.4769 | 0.3238 | 0.4140 |
| FR | AHIQ Lao et al. (2022) | 0.7138 | 0.5226 | 0.7554 | 0.7917 | 0.5935 | 0.7843 | 0.7462 | 0.5506 | 0.7434 | 0.7828 | 0.5810 | 0.7761 | 0.8124 | 0.6122 | 0.8149 |
| | CKDN Zheng et al. (2021) | 0.6883 | 0.5044 | 0.7448 | 0.7236 | 0.5217 | 0.7186 | 0.7024 | 0.5085 | 0.7073 | 0.6989 | 0.5030 | 0.6868 | 0.6997 | 0.5076 | 0.7045 |
| | DISTS Ding et al. (2020) | 0.6557 | 0.4749 | 0.6641 | 0.6385 | 0.4582 | 0.6405 | 0.6720 | 0.4856 | 0.7040 | 0.6875 | 0.4945 | 0.7040 | 0.7541 | 0.5540 | 0.7571 |
| | LPIPS Zhang et al. (2018a) | 0.6174 | 0.4401 | 0.6323 | 0.6742 | 0.4853 | 0.6405 | 0.6098 | 0.4374 | 0.5895 | 0.6188 | 0.4358 | 0.5678 | 0.6842 | 0.4844 | 0.6248 |
| | TOPIQ-FR Chen et al. (2024a) | 0.7876 | 0.5949 | 0.8192 | 0.8030 | 0.6011 | 0.8001 | 0.7944 | 0.5941 | 0.7898 | 0.8161 | 0.6111 | 0.8051 | 0.8084 | 0.6033 | 0.8160 |
| NR | CLIPIQA Wang et al. (2023) | 0.2597 | 0.1727 | 0.2715 | 0.3258 | 0.2201 | 0.3295 | 0.2608 | 0.1768 | 0.2473 | 0.3331 | 0.2252 | 0.3480 | 0.4283 | 0.2921 | 0.4228 |
| | CNNIQA Kang et al. (2014) | 0.3490 | 0.2356 | 0.3572 | 0.3909 | 0.2676 | 0.3737 | 0.5042 | 0.3497 | 0.4942 | 0.5044 | 0.3412 | 0.5092 | 0.6176 | 0.4311 | 0.6028 |
| | DBCNN Zhang et al. (2018b) | 0.5077 | 0.3527 | 0.5513 | 0.5553 | 0.3855 | 0.5580 | 0.6177 | 0.4259 | 0.6008 | 0.6373 | 0.4404 | 0.6217 | 0.6472 | 0.4496 | 0.6435 |
| | QualiClip Agnolucci et al. (2025) | 0.7072 | 0.5115 | 0.7188 | 0.7337 | 0.5366 | 0.7187 | 0.7027 | 0.5048 | 0.6866 | 0.7602 | 0.5537 | 0.7493 | 0.7352 | 0.5418 | 0.7379 |
| | TOPIQ-NR Chen et al. (2024a) | 0.7788 | 0.5818 | 0.8044 | 0.7974 | 0.6014 | 0.7993 | 0.7966 | 0.6011 | 0.7911 | 0.8071 | 0.6035 | 0.7967 | 0.7965 | 0.5952 | 0.8027 |

$R_3^6 = [r'_{ij}]$. The matrix $R_3^6$ can also be expressed as the product of rotations for joints 4, 5, 6 using their D-H parameters: $R_3^6 = \mathbf{R}_z(\theta_4)\mathbf{R}_x(\alpha_3)\mathbf{R}_z(\theta_5)\mathbf{R}_x(\alpha_4)\mathbf{R}_z(\theta_6)\mathbf{R}_x(\alpha_5)$. For the UR5 D-H parameters in Table 3: $R_3^6 = \mathbf{R}_z(\theta_4)\mathbf{R}_z(\theta_5)\mathbf{R}_x(\pi/2)\mathbf{R}_z(\theta_6)\mathbf{R}_x(-\pi/2)$. The symbolic product is:

$$R_3^6 = \begin{pmatrix} c_4 c_5 c_6 - s_4 s_5 c_6 & c_4 s_5 + s_4 c_5 & c_4 c_5 s_6 - s_4 s_5 s_6 \\ s_4 c_5 c_6 + c_4 s_5 c_6 & s_4 s_5 - c_4 c_5 & s_4 c_5 s_6 + c_4 s_5 s_6 \\ s_6 & 0 & -c_6 \end{pmatrix}. \tag{15}$$

By comparing elements of the numerically computed $R_3^6 = [r'_{ij}]$ with this symbolic form:

1. $r'_{32}$ must be 0. If the computed $((R_0^3)^\top R_0^6)_{32}$ is significantly non-zero, it indicates no solution for this wrist structure or a modeling error.

2. From $r'_{31} = s_6$ and $r'_{33} = -c_6$:
$$\theta_6 = \operatorname{atan2}(r'_{31}, -r'_{33}). \tag{16}$$
This provides a unique solution for $\theta_6$ in $(-\pi, \pi)$. Another solution is $\theta_6 \pm \pi$ (if $s_6, c_6$ are flipped), but usually we seek solutions within joint limits.

3. From $r'_{12} = c_4 s_5 + s_4 c_5 = s_{4+5}$ and $r'_{22} = s_4 s_5 - c_4 c_5 = -c_{4+5}$:
$$\theta_4 + \theta_5 = \operatorname{atan2}(r'_{12}, -r'_{22}). \tag{17}$$
Consider the movement of these two elements as a whole, we have $\phi_{45} = \theta_4 + \theta_5$.

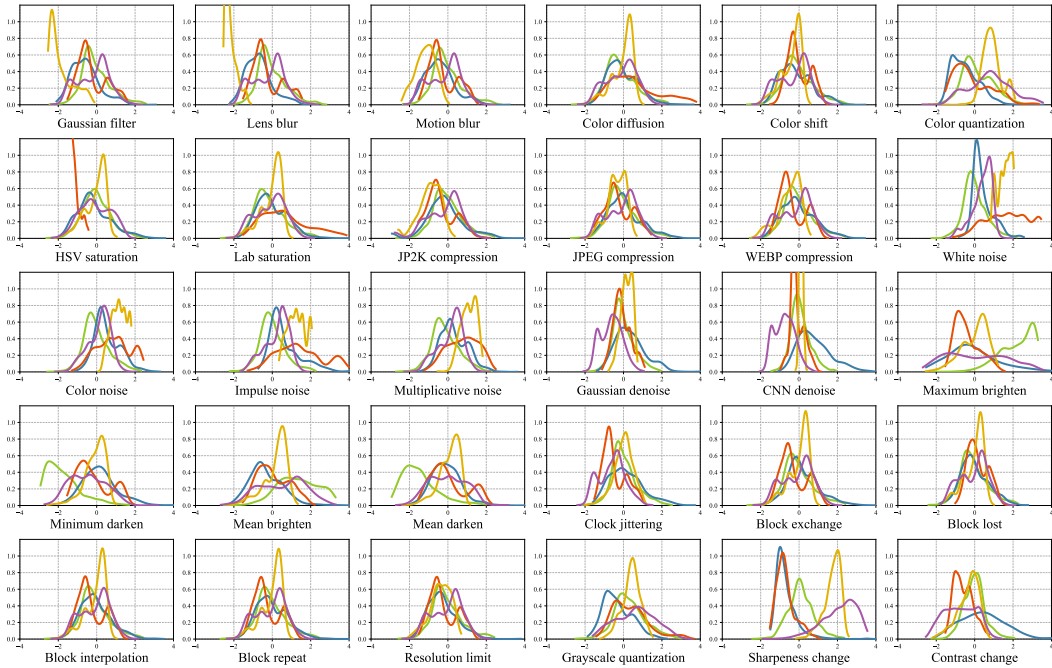

Figure 12: Low-level feature distribution of MPD, normalized and visualized in 30 corruption subsets. Different colors denote **Luminance**, **Contrast**, **Chrominance**, **Blur**, and **Spatial Information**.

4. To find $\theta_5$ and $\theta_4$ separately: $s_5 = c_4 r'_{12} + s_4 r'_{22} = c_4 s_{\phi_{45}} + s_4(-c_{\phi_{45}}) = \sin(\phi_{45} - \theta_4) = \sin\theta_5$. $c_5 = s_4 r'_{12} - c_4 r'_{22} = s_4 s_{\phi_{45}} - c_4(-c_{\phi_{45}}) = \cos(\phi_{45} - \theta_4) = \cos\theta_5$. A common method to solve for $\theta_5$ (wrist roll) for many spherical wrists involves:

$$\theta_5 = \sigma_5 \arccos\left( \frac{r'_{11} + r'_{22}\frac{s_4}{c_4}}{c_6(c_4 - s_4\frac{s_4}{c_4})} \right). \tag{18}$$

According to $\theta_5$ we have $\theta_4 = \phi_{45} - \theta_5$. Thus all rotation angles can be retrieved.

Typically, UR5 has 8 unique inverse kinematics solutions ($\sigma_1 = \pm 1, \sigma_3 = \pm 1, \sigma_5 = \pm 1$ for the choice of $s_5$). Singularities (e.g., $s_5 = 0$) lead to infinite solutions where $\theta_4$ and $\theta_6$ are coupled.

### E.5 HANDLING SINGULARITIES

- Shoulder Singularity: Occurs if $x_{wc}^2 + y_{wc}^2 - (d_4 + d_5)^2 = 0$. The wrist center lies on the $z_0$ axis (for $d_4 + d_5 = 0$) or a cylinder around $z_0$. $\theta_1$ is not uniquely defined.
- Elbow Singularity: Occurs if $K_x^2 + K_z^2 - a_2^2 - a_3^2 = \pm 2a_2 a_3$, meaning $\cos(\theta_3) = \pm 1$ (arm fully extended or folded). $\sin(\theta_3) = 0$, so $\theta_2$ solution becomes simpler but an infinite number of $\theta_2$ might exist if $p_{wc}$ is on $z_1$.
- Wrist Singularity: Occurs if $s_5 = 0$ (i.e., $\theta_5 = 0$ or $\pi$). Axes $z_4$ and $z_6$ align. In this case, only the sum or difference ($\theta_4 \pm \theta_6$) can be determined according to Section D.4. One angle can be chosen arbitrarily, and the other is then fixed.

## F SUBJECTIVE PERCEPTION TASK DEFINITION

Before VLM and VLA inference, we organized five Ph. D. candidates as a panel to define five downstream tasks for each image as shown in Figure 10. To avoid bias from a single subject, each sample is sent to five subjects in a random order to design specific tasks based on the image. The 1,230 samples to be annotated come from seven Robotic database Khazatsky et al. (2024); Kalashnikov et al. (2018); Kerr et al. (2023); Depierre et al. (2018); Tziafas et al. (2023); Gu et al. (2023); Rosete-Beas

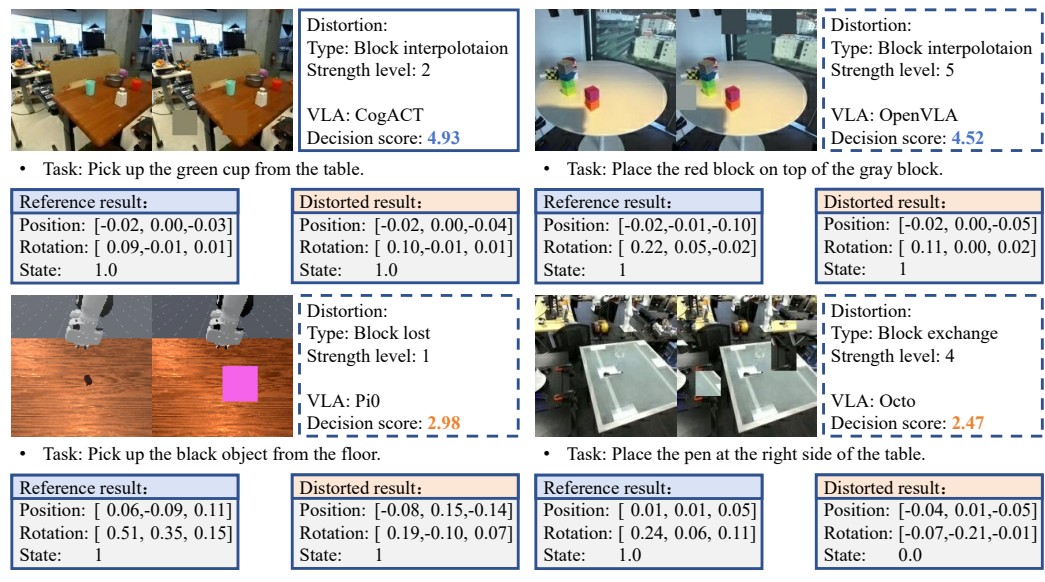

Figure 13: Positive and negative cases. Slight distortion may significantly affect the inference result of Embodied AI, while severe distortion may not. Emphasizing the significance of Embodied IQA.

et al. (2022), and are filtered according to the settings in the main text, to retain only high-quality images. Subjects can see the previous tasks, and the new tasks they design need to be more difficult than them and test different abilities (for example, if an object has been pushed, try not to grab it again). Images with display errors, NSFW, or offensive content will be removed. After each image has five task labels, a professional robotics engineer will adjust the specific samples. Based on operational experience, the difficulty of the five tasks will be re-ranked, and unreasonable tasks will be modified.

## G   COGNITION IQA EXPERIMENT

Due to space limitations in the main text, we mainly discuss the Decision step (specific to Embodied AI), and the Cognition step (common to general machines) is listed in this section. First, the Cognition score given by VLM is shown in Figure 11. Compared with the three dimensions of Decision in Figure 6, the correlation between the Cognition dimensions is higher, and the distribution difference between different categories of data is smaller. This fully demonstrates the difference between the reasoning mechanisms of VLM and VLA, and proves the rationality of separating these two steps.

Therefore, in addition to Decision, we also conducted IQA experiments on Cognition, following the training/testing settings in the main text. Table 4 presents the performance of advanced quality metrics on Cognition, compared with Decision in Table 2, current IQA metrics has better prediction results on Cognition. Since the current IQA method is more related to VLM than VLA, the quality indicators that general machines already have are initially available, but Embodied AI cannot be effectively evaluated. It is worth mentioning that the zero-shot baseline method on Cognition can occasionally even achieve an SRCC of more than 0.7, surpassing a number of fine-tuned methods; while the baseline on Decision is significantly weaker. This is exactly why we separated the Robot Visual System from the Machine Visual System and used the Mortonian system to model the Intelligent System in four steps. In short, we hope that the Embodied IQA database can promote more complete quality indicators, whether applied for VLM or VLA as subjects in Embodied tasks.

## H   LOW-LEVEL ATTRIBUTE DISTRIBUTION

Figure 12 shows the distribution of low-level features of all instances of Embodied IQA. After overall regularization, 30 types of corruption are grouped and displayed. The features considered include Luminance, Contrast, Chrominance, Blur, and Spatial Information. There are significant differences

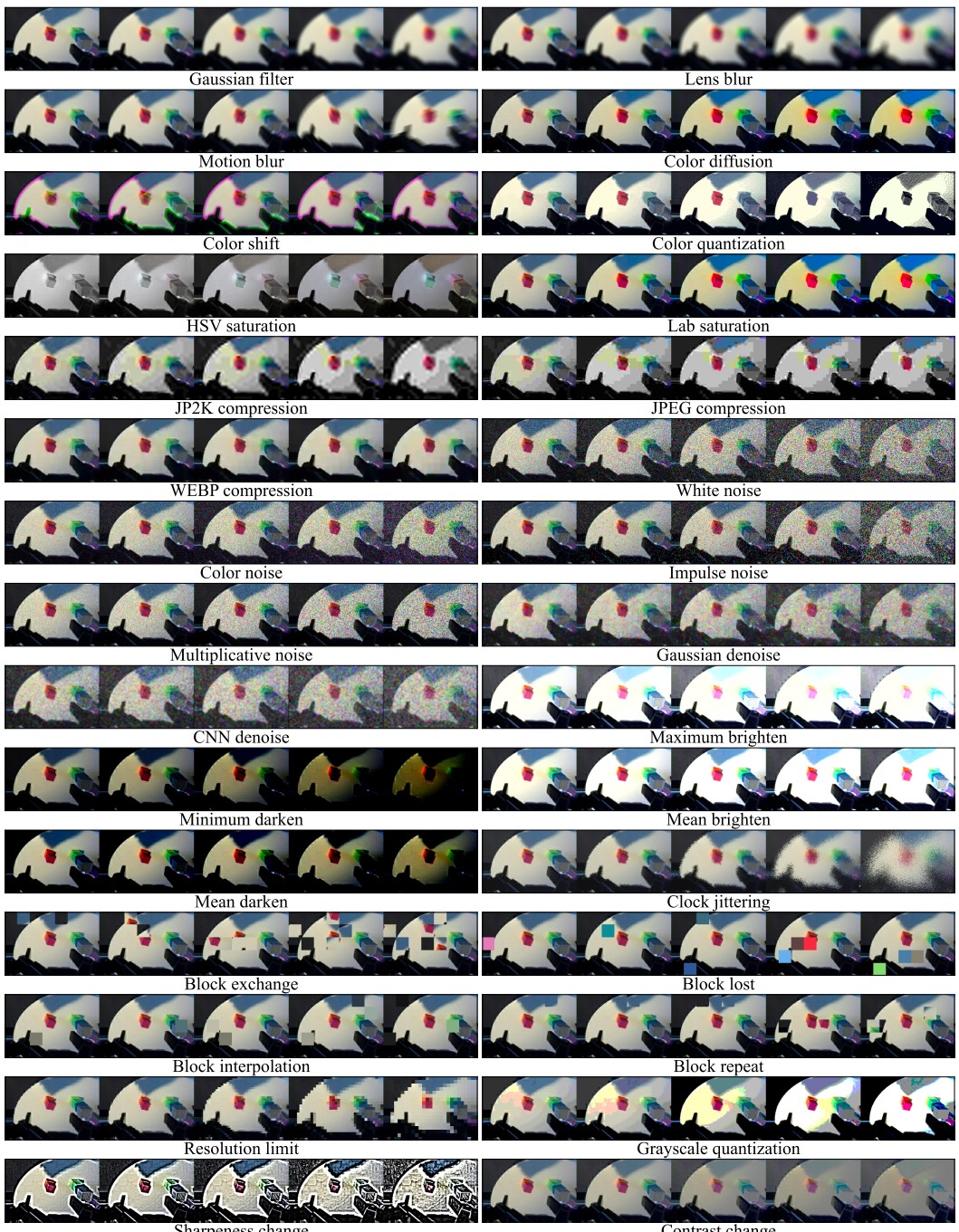

Figure 14: Visualization of 30 distortion types. Strength from left (Level 1) to right (Level 5).

in these low-level attributes for different corruptions. For example, in the first three blurry cases, the blur curve is left-biased and becomes right-biased after sharpening. In general, similar corruption categories will lead to similar results (such as five noise-related and four block-related). Two of the denoises have the sharpest distributions; Color quantization, Grayscale quantization, Sharpness change, and Contrast change are the most irregular. These findings deserve further exploration.

## I  CASES STUDY

Figure 13 shows four typical examples from the Embodied IQA database (center-cropped for visualization), including the VLA inference results for reference/distorted image pairs under different distortions. The results in the upper left and lower right corners are as expected, the more severe the distortion, the lower the score. However, for the distortion level 5 in the upper righ, since it does not affect any objects on the desktop, the 'gray block' as the target of the task is not affected, so the subjective score is as high as 4.52; on the contrary, although the distortion level in the lower left corner is only 1, the 'lost macro block' happens to be the target object, so the VLA Position and Rotation are greatly changed with a score only 2.98. Figure 14 shows 30 distortion types at different strength levels from 1 to 5. In the previous human-oriented scenario, the visual quality of different corruptions is similar at the same strength. However, from the example above, the preference of Embodied AI depends on the task, which significantly differs from traditional IQA paradigm. We hope that our database can further inspire better quality metrics for Embodied AI.

## J  DISCLAIMER

The main purpose of this study is to apply IQA to Embodied AI to promote its Real-world application, rather than to praise or criticize any VLM, VLA, or IQA model. We evaluate image samples rather than models. Lower scores do not mean that the performance of downstream VLM/VLA is poor, but distortion has a greater impact on it; similarly, lower correlation coefficients do not mean defects in the IQA method, but rather indicate the huge difference between Embodied and traditional IQA. Considering the scale of the database, we will open it in several stages for non-commercial use, and sincerely hope that future robotic-oriented IQA metrics can drive the development of Embodied AI.

