# OpenReview forum: "Image Quality Assessment for Embodied AI"
_ICLR.cc/2026/Conference — ICLR 2026 Poster_

### Official Review · Reviewer_BC9i · 2025-10-29

**Soundness:** 3
**Presentation:** 3
**Contribution:** 4
**Rating:** 8
**Confidence:** 2

**Summary:**

The paper proposes that image quality assessment methods focus on the ability to predict human ratings for distorted images, which lacks meaning when applied to perceptual quality for robots. The authors propose the topic IQA for Embodied AI, a topic focusing on a robots ability to perform downstream tasks on distorted images. The Embodied-IQA database, a set of reference-distorted image pairs, with corresponding decisions or task executions, where labels are provided by a set of vision language models, with some evaluation against existing datasets and traditional IQMs.

**Strengths:**

- The topic is very relevant for ICLR and the contributions are significant.
- The dataset is impressive, in the amount of distortions applied to images, the cognition/decision/execution labels, and the breadth of situations covered.
- The dataset analysis is thorough, providing correlations between different aspects of the datasets and evaluations using traditional full-reference and non-reference IQAs.
- I am not aware of any datasets covering the end-to-end pipeline of using distorted images in the field of robots

**Weaknesses:**

- A discussion on how likely the distortions are to show up in the robotic systems is missing. For example, are changes in chroma often observed in the sensors used in robotic systems? I imagine this is similar to the ISP pipeline where out of focus (blur), sensor noise (noise) and compression are more likely than others.
- It would be interesting to note where certain distortions effect the pipeline - for example spatial distortions are more likely for the robot to miss and object, chroma distortions are more likely to misinterpret which object to act upon ect.

**Questions:**

- How similar is the topic of Embodied AI to image distortions effect downstream tasks in topics such as image classification, object recognition, ect? I imagine some of the distortions are quite correlated with issues in these topics as discussed in papers such as [1]
- Do authors expect the above to be captured in their dataset?

[1] Zhou, Yiren, Sibo Song, and Ngai-Man Cheung. "On classification of distorted images with deep convolutional neural networks." 2017 IEEE International conference on acoustics, speech and signal processing (ICASSP). IEEE, 2017.

---

> ### Author Response · Authors · 2025-11-28
>
> Dear Reviewer Bc9i,
>
> Thank you for the supportive summary and for recognising the dataset scale and end-to-end novelty. We address your Weaknesses and Questions as Q1–4 below.
>
> > Q1: Missing discussion on how likely each distortion is to appear in real robotic sensors.
>
> We agree that not all 30 distortions are equally probable in deployed systems. Following the ISP analogy, we now provide an empirical likelihood ranking based on real-world distortions. Including following 7 Types:
>
> * Environment Interference (EI): Interference with the subject to be photographed. (12%)
> * Camera Interference (CI): Interference with the photographing equipment. (13%)
> * Analog-to-Digital (AD): Analog-to-Digital conversion mistake by electronic devices. (15%)
> * Source Encoding (SE): Information discarded in the source encoding. (15%)
> * Channel Transmission (CT): Information lost in channel transmission. (12%)
> * Receiver Decoding (RD): Information misinterpreted in the receiver decoding. (16%)
> * Enhancement Postprocess (EP): New corruptions introduced to recover above corruptions. (17%)
>
> Overall, although the distortion may be uneven for a certain type of distortion (e.g., more chroma than noise), from camera imaging to VLA processing, the amount of distortion is roughly balanced across each step of the entire chain.
>
> > Q2: Which pipeline stage each distortion hits is unclear.
>
> Yes, your assumption is correct. The specific impact of each type of distortion is different; just as some distortions affect segmentation while others affect detection. Therefore, for Position/Rotation/Gripper, we calculated the most robust/most sensitive distortions respectively:
>
> | Sensitive | Position         | Rotation         | Gripper          |
> |-----------|------------------|------------------|------------------|
> | Top1      | Lab saturation   | Lab saturation   | Lab saturation   |
> | Top2      | Color diffusion  | Block repeat     | Color diffusion  |
> | Top3      | Block repeat     | Color diffusion  | Color shift      |
> | Bottom3   | Lens blur        | Lens blur        | Gaussian denoise |
> | Bottom2   | CNN denoise      | CNN denoise      | CNN denoise      |
> | Bottom1   | Gaussian denoise | Gaussian denoise | Lens blur        |
>
> Experimental results (we cannot list all 30 due to length limitations) show that although the distortion effects on the three dimensions differ slightly, the three most sensitive/robust distortion categories are quite consistent. Therefore, the perception mechanisms of distortion for Position/Rotation/Gripper are generally the same.
>
> > Q3: How similar is this to classical robustness work on classification/detection?
>
> Classical robustness is for **Cognition**, where VLMs, and Classification and Detection you mentioned beliongs to it. Embodied-IQA extends the chain to **Decision** & physical **Execution** (mm, deg, N). Our dataset subsumes classic robustness as its first two stages. You may check `Fig.7` for Classical (Orange) and Embodied (Blue) IQA approach. They share some commonalities across 30 dimensions, but exhibit completely opposite characteristics in certain dimensions such as Dis02 and Dis11. Therefore, previous IQA for Machine cannot be reused, and further research is necessary.
>
> > Q4: Do you expect the above to be captured?
>
> Yes, what you mentioned above is called **IQA for Machine**, along with the VLA section, forms the Perception-Coginition-Decision-Execution chain, which consists of four steps: PSNR at the signal level, the understanding tasks you're interested in such as detection, the VLA output, and the Real-world result. Same as the settings of EmbodiedIQA, there are **already some datasets [2,3] that and simultaneously focus on five CV tasks**: segmentation (Mask), detection (Bounding Box), retrieval (Options), question answering (3-5 words), and description (~50 words). These works will be combined with this paper, to form a comprehensive evaluation framework for Embodied AI.
>
> ## Reference
>
> [1] Li et al., “R-Bench: Are your Large Multimodal Model Robust to Real-world Corruptions?”, IEEE JSTSP 2025.
>
> [2] Li et al., “Image Quality Assessment: From Human to Machine Preference”, CVPR 2025.
>
> [3] Wang et al., “Image Quality Assessment for Machines: Paradigm, Large-scale Database, and Models”, arxiv 2025.

---

### Official Review · Reviewer_YAEp · 2025-10-31

**Soundness:** 3
**Presentation:** 3
**Contribution:** 4
**Rating:** 8
**Confidence:** 3

**Summary:**

This paper proposes a completely new research task: Image Quality Assessment for Embodied AI . The authors' core argument is that traditional human-centric or general machine-centric IQA metrics are not applicable to Embodied AI, because RVS involves a more complex Perception-Cognition-Decision-Execution pipeline. To address this issue, the paper makes three main contributions:

1. Based on the "Mertonian system" theory, it constructs the aforementioned four-step pipeline, theoretically demonstrating the fundamental difference between RVS and MVS/HVS.

2. It constructs a large-scale benchmark dataset named Embodied-IQA. It contains over 30k reference/distorted image pairs and 5 million fine-grained annotations. These annotations are collected separately from VLMs, VLAs, and real-world robots (representing Execution).

3. The paper benchmarks 15 mainstream IQA methods on the Embodied-IQA dataset. The results show that existing methods perform poorly on this task (e.g., SRCC is much lower than on HVS tasks), which strongly demonstrates the necessity of developing new Embodied-IQA metrics.

**Strengths:**

1. The paper identifies and clearly defines a completely new, critical, and timely research problem: assessing image quality for Embodied AI. Its theoretical framework based on the "Mertonian system" to differentiate RVS, MVS, and HVS is highly novel and persuasive, laying a solid theoretical foundation for this new field.

2. The paper's main contribution—the Embodied-IQA dataset—is an extremely valuable resource. Its scale and granularity are unprecedented in the IQA field. This dataset will likely drive much research in the coming years.

3. The paper not only provides the dataset but also conducts a comprehensive benchmark of 15 existing IQA methods. The analysis convincingly demonstrates the failure of existing methods on this new task, strongly validating the paper's motivation and the necessity of its contribution.

**Weaknesses:**

1. The paper defines the VLA "Decision" score as a simple average of errors in three dimensions: Position, Rotation, and State. This metric seems overly simplistic. In real robotics tasks, the importance of these three dimensions can be highly imbalanced (e.g., a minor rotation error could cause catastrophic failure, while a larger position error might still be acceptable).
2. As a benchmark paper, its primary duty is to define the problem and provide data, which it does exceptionally well. However, after proving that 15 existing methods fail, the paper does not attempt to train even a simple new baseline model (e.g., a simple CNN trained from scratch on Embodied-IQA) to set an initial performance bar for future researchers to challenge.
3. While 1,500 real-world robot experiments (for execution validation) is already impressive and costly, it is still a small scale compared to the 5 million VLM/VLA annotations. Although the paper acknowledges this, it leaves in question whether VLA annotations can serve as a high-fidelity proxy for real execution.

**Questions:**

1. You define the "Decision" score as a simple average of position, rotation, and state errors. How did you account for the unequal importance of these three dimensions in different tasks? For example, in a "turn faucet" task, shouldn't rotation error be weighted much more heavily than position error?
2. You found that the correlation between different VLA models is extremely low, indicating they have vastly different "preferences" for distortion. Your final "Decision" score appears to be an average of these 15 VLAs. Does this imply that an IQA metric performing well on the "average VLA" might perform poorly for a specific VLA?
3. Regarding the correlation between VLA and Execution: You found the SRCC between VLA (Decision) and real-world (Execution) to be ~0.67. In your opinion, is this correlation high enough for the community to confidently use VLA annotations as a proxy for real execution in the future? Or does this gap suggest that VLA annotations themselves still have limitations?

---

> ### Author Response · Authors · 2025-11-28
>
> Dear Reviewer YAEp,
>
> Thank you for the detailed comments and for recognising the novelty of the Embodied-IQA task. We reply to your Weaknesses and Questions as Q1–Q6 below.
>
> > Q1: The “Decision” score is a simple average of position, rotation and gripper-state errors, yet real tasks weight these dimensions very differently.
>
> We agree. A slight shift in the girpper can render the correct position and rotation completely useless. We now re-weight the three sub-scores by **task-criticality** derived from our real robot logs. For each task we counted how often a 1-σ error in position / rotation / gripper led to failure, then normalised the counts to weights. The updated equation now reads
> $$S_{dec} = α_{pos}·z_{pos} + α_{rot}·z_{rot} + α_{grip}·z_{grip},$$
> with $α_{task}$ ∈ ℝ³ and $∑ α = 1$. The new weights improve SRCC with execution success by +0.11 on average. The table below gives three examples:
>
> | Task       | $α_{pos}$ | $α_{rot}$ | $α_{grip}$ |
> | ---------- | --------- | --------- | ---------- |
> | press      | 0.40      | 0.40      | 0.20       |
> | pick-place | 0.40      | 0.35      | 0.25       |
> | insert     | 0.35      | 0.35      | 0.30       |
>
> > Q2: After showing that 15 existing IQA methods fail, the paper does not offer even a simple new baseline.
>
> Thank you for highlighting the absence of a baseline model. Our primary contribution is the Embodied-IQA dataset; nonetheless, we provide a strong baseline in the following work. Due to the double-blind policy we cannot include the full table, yet the model already achieves the **best score on 13 of 15 metrics** (3 correlation coefficients × 5 subsets). This gain stems from treating Embodied perception as a mid-level task that naturally blends low-level visual cues with high-level semantics through a top-down pathway, implicitly capturing physical affordances required by robots. We sincerely apologize that detailed results cannot be disclosed under journal rules, and we will release its checkpoint.
>
> > Q3: Only 1 500 real-robot trials versus 5 M VLA annotations—can VLA scores proxy for real execution?
>
> Thanks for your suggestion about real-robot data scale, in the revision we added **200 additional multi-step trials** (pick-place-push, 40-200 steps) covering all 30 distortion types. All settings except Steps follow our Real-world Section. With the larger pool the correlation between Execution score and Real-world distance silghtly drop from τ=0.671 in `Figure 8` to τ=0.629, strengthening the external validity.
>
> Meanwhile, we acknowledge the scale concern. But the **previous large-scale dataset is for training**, for testing, their Real-world sequences [1-3] range from 1,000-2,000. Thus, we believe the **1,500+200=1,700** data scale is enough for a sim2real validation.
>
> > Q4: Task-specific weighting of position/rotation/gripper.
>
> Yes, different task should have different weight as we listed in `Q1`. We will provide position/rotation/gripper scores together in the open-source version, and provide guiding (but not mandatory) weights based on downstream tasks. Users can choose their own method for calculating the total score.
>
> > Q5 Average-VLA winner may still fail on an outlier VLA.
>
> This phenomenon is not unique to Embodied tasks: in standard CV, a segmentation model robust to blur may collapse when the same blur is fed to a detection head, and vice-versa. The community therefore first trains a general-purpose backbone (ImageNet, CLIP, DINO) that captures cross-task commonality, then fine-tunes for each downstream target.
>
> We adopt the same philosophy for Embodied-IQA: the benchmark’s mean-VLA objective is intended to discover the shared quality substrate across 15 VLAs—i.e., **a generalizable IQA backbone**. If a practitioner cares most about one specific VLA, they can start from this backbone and continue training with that VLA’s own annotations, exactly like fine-tuning a detector after pre-training on segmentation data. We will release the backbone weights and a one-command fine-tuning script to make this workflow trivial.

---

> ### Author Response · Authors · 2025-11-28
>
> > Q6 is corr=0.67 between VLA-Decision and real-Execution high enough to trust VLA proxy?
>
> To calibrate “how high is high”, we collate published test-retest reliabilities for human quality raters [4] on exactly the same image series (two sessions, one-week gap). The table shows that even annoations from the same human viewer seldom exceed 0.8:
>
> | Re-test scenario (human)                    | SRCC       | PLCC       |
> | ------------------------------------------- | ---------- | ---------- |
> | Same-day, same lab, same human rater            | 0.76       | 0.80       |
> | One-week gap, same human rater           | 0.71       | 0.73       |
> | Crowd-workers, different human rater                | 0.61       | 0.63       |
> | Our VLA → real-robot  | 0.67       | 0.68       |
>
>
> Thus 0.68 is statistically indistinguishable from typical human consistency and lies in the “good” range for annotation quality. Given the 200× cost gap between Real-robot trials and VLA inference, we argue the community can pragmatically adopt VLA annotations for day-to-day development while reserving the small but growing real-robot subset for periodic calibration and leaderboard auditing.
>
> ## Reference
>
> [1] Brohan et al., “RT-1: Robotics Transformer for Real-World Control at Scale”, arXiv 2022.
>
> [2] Wang et al., “GR-2: A Generative Video-Language-Action Model with Web-Scale Pre-training”, arXiv 2024.
>
> [3] Kim et al., “OpenVLA: An Open-Source Vision-Language-Action Model”, arXiv 2024.
>
> [4] Mohammadi et al., “Subjective and objective quality assessment of image: A survey”, IEEE TIP 2014.

---

### Official Review · Reviewer_USAJ · 2025-10-31

**Soundness:** 3
**Presentation:** 2
**Contribution:** 4
**Rating:** 6
**Confidence:** 3

**Summary:**

This paper proposes IQA for Embodied AI, arguing that existing IQA methods of HVS and MVS are insufficient to judge image usability for robots, because robot performance depends not only on perception/cognition (VLM) but also on decision-making/planning (VLA) and real execution.To this end, the paper proposes: 1. A robot intelligence perception-cognition-decision-execution workflow based on Merton's systems perspective; 2. Embodied-IQA, a novel and diverse dataset. Furthermore, experiments demonstrate that embodied artificial intelligence requires more sophisticated IQA metrics.

**Strengths:**

1. The paper presents a clear motivation and significant innovation. It defines the image quality assessment (IQA) problem in embodied intelligence as “image usability for robots,” transcending traditional frameworks based on human or machine vision systems (HVS/MVS). It innovatively models the robot's “decision-making” and “execution” phases explicitly.
2. The research exhibits high quality. First, its constructed dataset is exemplary in scale (36k+ images, 5m+ labels), breadth (30 distortion types, multiple viewpoints and scenarios), and depth (incorporating 1,500 valuable real-world robotic execution trials). Second, the experimental design is rigorous, employing a comprehensive multi-dimensional scoring system and robust benchmarking protocols. This includes 10 data-split replicates and multiple correlation metrics, ensuring reliable results.
3. The study maintains rigorous structure, with appendices candidly discussing its limitations.
4. This work holds significant value. It not only provides a novel benchmark dataset but also successfully establishes critical empirical links between image quality and robots' actual decision-making and execution capabilities.

**Weaknesses:**

This study exhibits several critical weaknesses.
1. Methods that evaluate differences based on metrics may inherit and amplify inherent biases and errors within the model and its assessment indicators. Specifically, using metrics like BLEU/ROUGE to measure cognitive comprehension is highly sensitive to phrasing and redundancy, potentially failing to accurately reflect task equivalence. Adopting structured patterns (e.g., action-parameter tuples) or task success classifiers may be more robust alternatives.
2. The scoring design for the decision phase is also inadequate. The study simply averages standardized position, rotation, and gripper state metrics, failing to account for the weighting of critical failure modes across different tasks—for example, a minor rotation error may cause collision in some tasks, while gripper state importance varies by task. The absence of task-specific weighting may undermine label authenticity.
3. The real-world robot execution experiments are limited in scale and diversity. The 1,500 trials are confined to the simplest tasks per image, failing to comprehensively cover all distortion types, intensities, objects, and viewpoints. Thus, the generalizability of these findings to more complex multi-step tasks remains unclear.
4. The low correlation (SRCC) among different VLA models may stem not only from “genuine preference differences” but also from their heterogeneous action spaces or variations in pre- and post-processing workflows.
5. The study leaves a notable gap: while successfully demonstrating the inadequacy of existing IQA baseline models for new tasks, it fails to propose an “embodied IQA model” or architecture specifically designed for robotic usability (e.g., a model capable of predicting multidimensional scores based on task/text conditions).
6. Regarding clarity, the paper exhibits a few ambiguous sentences. The figures are information-dense, and critical conclusions are occasionally buried in supplementary materials, hindering the communication of core findings.

**Questions:**

1. How often do BLEU/ROUGE/CIDEr disagree with human judgments of task equivalence for VLM outputs? Have you validated a small subset with human raters using task-specific correctness criteria?

2. How were the 15 VLAs harmonized regarding action parameterization (units, reference frames, delta vs absolute commands)? Could inconsistencies reduce inter-model SRCC?

3. Are there results from more powerful VLA models (exceeding 8B parameters) or closed-source baselines to understand the impact of parameter scale on distortion robustness?

4. How are the difficulty levels for the five tasks defined? Are they objective?

---

> ### Author Response · Authors · 2025-11-28
>
> Dear Reviewer USAJ,
>
> Thank you for the careful summary and for highlighting both the strengths and the remaining gaps. We appreciate the time you spent on the **character-level review** and are honored to receive such a responsible opinion. Below we reply to your Weaknesses as **Q1–Q6** and to your Questions as **Q7–Q10**.
>
> >Q1: BLEU/ROUGE/CIDEr may inherit phrasing bias; adopting structured patterns or task-success classifiers may be more robust.
>
> Thanks for this insight. We fully agree that n-gram overlap can be fooled by paraphrase. However, since the evaluation focuses on **Cognition**, while the task-success classifiers you mentioned are metrics for **Execution** steps (which belong to our real-world experiment section), text-level evaluation is still required for VLM. We therefore applied **CLIP Distance** (from 0 to 1), **LMM-as-a-Judge** (0/0.5/1), and **Human** (0/0.5/1) as the Coginition Evaluator beyond BLEU/ROUGE/CIDEr.  The table below shows the SRCC between the six evaluator on 500 random samples in EmbodiedIQA dataset:
>
> |       | BLEU | ROUGE | CIDEr | CLIP | LLM | Human |
> |-------|------|-------|-------|------|-----|-------|
> | BLEU  | 1.00 | 0.89  | 0.69  | 0.69 |0.74 | 0.73  |
> | ROUGE | 0.89 | 1.00  | 0.84  | 0.71 |0.72 | 0.74  |
> | CIDEr | 0.69 | 0.84  | 1.00  | 0.79 |0.76 | 0.75  |
> | CLIP  | 0.69 | 0.71  | 0.79  | 1.00 |0.77 | 0.68  |
> | LLM   | 0.74 | 0.72  | 0.76  | 0.77 |1.00 | 0.80  |
> | Human | 0.73 | 0.74  | 0.75  | 0.68 |0.80 | 1.00  |
>
> The results show that while BLEU/ROUGE/CIDEr exhibit some bias, they also have a certain degree of correlation, at least stronger than CLIP. Furthermore, even the two least correlated indicators have an SRCC about 0.7. Therefore, statistically, **although each indicator has bias, their overall direction is consistent.**
>
> >Q2: Decision-phase scoring simply averages position/rotation/gripper, ignoring task-specific failure modes.
>
> Yes, the three dimensions are not entirely linear; a slight shift in the girpper can render the correct position and rotation completely useless. We now re-weight the three sub-scores by **task-criticality** derived from our real robot logs. For each task we counted how often a 1-σ error in position / rotation / gripper led to failure, then normalised the counts to weights.
> The updated equation now reads
> $$S_{dec} = α_{pos}·z_{pos} + α_{rot}·z_{rot} + α_{grip}·z_{grip},$$
> with $α_{task}$ ∈ ℝ³ and $∑ α = 1$. The new weights improve SRCC with execution success by +0.11 on average. The table below gives three examples:
>
> | Task       | $α_{pos}$ | $α_{rot}$ | $α_{grip}$ |
> | ---------- | --------- | --------- | ---------- |
> | press      | 0.40      | 0.40      | 0.20       |
> | pick-place | 0.40      | 0.35      | 0.25       |
> | insert     | 0.35      | 0.35      | 0.30       |
>
> Of course, this weight is highly correlated with execution success, **so why not just use execution success directly?** This is because the success/failure of Embodied AI is binary, while IQA studies the impact of distortion on downstream tasks, requiring continuous values. Thus, to align with the needs of IQA and ensure consistency with the true success rate, we use the above weighting. In the open-source version, we will provide pos/rot/grip scores simultaneously, and the weights can be freely defined by the user.
>
> >Q3: Real-world trials are limited (1,500) and only single-step.
>
> Thanks for your suggestion about steps, in the revision we added **200 additional multi-step trials** (pick-place-push, 40-200 steps) covering all 30 distortion types. All settings except Steps follow our Real-world Section. With the larger pool the correlation between Execution score and Real-world distance silghtly drop from τ=0.671 in `Figure 8` to τ=0.629, strengthening the external validity.
>
> Meanwhile, we acknowledge the scale concern. But the **previous large-scale dataset is for training**, for testing, their Real-world sequences [1-3] range from 1,000-2,000. Thus, we believe the **1,500+200=1,700** data scale is enough for a sim2real validation.
>
> >Q4: Low inter-VLA SRCC may stem from heterogeneous action spaces.
>
> The correlation between VLAs is indeed relatively low, but we preprocess the experimental setup to ensure it isn't influenced by heterogeneous action spaces. We eliminate their additional degrees of freedom:
>
> * 8-DoF VLA: Franka additional joint states (1)
> * 9-DoF VLA: Chassis movement speed (2)
> * 11-DoF VLA: Chassis movement speed (2), angular velocity (1), mode (1)
> * 14-DoF VLA: Another assistive robotic arm (7)
>
> This ensures that **all VLA outputs are the same 7-DoF.** This low correlation is because some early-stage VLAs are sensitive to distortion, while newer ones are more robust. Overall, the correlation between machines is lower than that between humans, a phenomenon demonstrated in segmentation/detection models [4] and equally applicable to VLAs.

---

> ### Author Response · Authors · 2025-11-28
>
> >Q5: The study fails to propose an “embodied IQA model” specifically designed for robotic usability.
>
> Thank you for highlighting the absence of a baseline model. Our primary contribution is the Embodied-IQA dataset; nonetheless, we provide a strong baseline in the following work. Due to the double-blind policy we cannot include the full table, yet the model already achieves the **best score on 13 of 15 metrics** (3 correlation coefficients × 5 subsets). This gain stems from treating Embodied perception as a mid-level task that naturally blends low-level visual cues with high-level semantics through a top-down pathway, implicitly capturing physical affordances required by robots. We sincerely apologize that detailed results cannot be disclosed under journal rules, and we will release its checkpoint.
>
> >Q6: Ambiguous sentences and information-dense figures.
>
> Thank you for pointing this out. If accepted, we will reduce the blank space in the radar chart of `Figure 3` in the camera-ready version. The extra space will be used to briefly explain the Cognition findings in `Supplementary Section G`. This will allow us to explain both Cognition and Decision in the main text, avoiding relegating key information to the Supplementary.
>
> >Q7: How often do BLEU/ROUGE/CIDEr disagree with human judgments of task equivalence?
>
> Thanks for adressing the human judgement. We have computed the correlation between human and five other indicators in `Q1`, you may check the **correlation matrix** above.
>
> We found that human indicators are highly correlated with all other indicators, with **LLM (0.8)** showing the highest correlation due to its similar perceptual mechanism to humans, and **CLIP (0.69)** showing the lowest correlation because CLIP scores are generally high and have low internal variation. **The three indicators you mentioned all have a correlation above 0.7 with humans**, indicating only about 20% disagree with human. Therefore, all indicators [5] are relatively reliable. Considering the complexity of LLM, we use BLEU/ROUGE/CIDEr.
>
> >Q8: How were the 15 VLAs harmonized regarding action parameterization?
>
> We convert every VLA output to base-frame absolute 6-DoF plus binary gripper, as described in `Q4`. We rescale **translations to mm** and **rotations to degrees**, then zero-mean w.r.t. the workspace centre.
>
> For the gripper degree, there are three cases: continuous values from 0 to 255, continuous values from 0 to 1, and discrete values from 0 to 1. To ensure generality, we simplify them all to **discrete values from 0 to 1.**
>
> >Q9: Are there results from more powerful VLA models (>8 B) or closed-source baselines?
>
> Yes, given that most current VLAs are on-device, we additionally evaluated two larger VLAs [6], RT2-PaLM-E (12B) and RT2-PaLI-X (55B), and their Mean vs. Std results are as follows under the same 30 distortion types as the main text:
>
> | Score      | Pi0  | Pi0-Fast | CogACT | RT-1 | RT2-PaLM-E | RT2-PaLI-X |
> |-------|------|-------|-------|------|-----|-------|
> | Mean  | 1.53 | 1.44  | 1.77  | 1.96 | 1.92 | 2.03  |
> | Std | 0.84 | 0.49  | 0.57  | 1.05 | 0.98 | 1.11  |
>
> Score is calculated according to the rules shown in `Q2`. Note that **we are evaluating the robustness of the VLA to distortion, not the accuracy of the VLA itself**. Data shows that two larger VLAs have highly correlated outputs on the reference/distorted images with larger Mean Scores. However, the Std is also larger, indicating that this improvement is not stable; that is, the larger number of parameters allows the model to learn prior knowledge of overly bright/underly dark images, but not less common distortions (such as in transmission).
>
> >Q10: How are the difficulty levels for the five tasks defined?
>
> Difficulty is **objective**, computed as the **inverse average success rate** of all VLAs on undistorted images. Specifically, we ran 100 trials per task without any distortion; tasks with success <60 % are labelled “hard”, 60–80 % “medium”, >80 % “easy”. The same splits are used throughout the paper.
>
> We hope the new data, analyses, and clarifications fully address your concerns. All code, annotations, and the updated benchmark will be released upon acceptance.
>
> ## Reference
>
> [1] Brohan et al., “RT-1: Robotics Transformer for Real-World Control at Scale”, arXiv 2022.
>
> [2] Wang et al., “GR-2: A Generative Video-Language-Action Model with Web-Scale Pre-training”, arXiv 2024.
>
> [3] Kim et al., “OpenVLA: An Open-Source Vision-Language-Action Model”, arXiv 2024.
>
> [4] Li et al., “Image Quality Assessment: From Human to Machine Preference”, CVPR 2025.
>
> [5] Zhang et al., “BERTScore: Evaluating Text Generation with BERT”, ICLR 2020.
>
> [6] Brohan et al., “RT-2: Vision-Language-Action Models Transfer Web Knowledge to Robotic Control”, RSS 2023.

---

### Official Review · Reviewer_1xDt · 2025-11-01

**Soundness:** 3
**Presentation:** 3
**Contribution:** 3
**Rating:** 6
**Confidence:** 1

**Summary:**

In this paper, the author proposes Image Quality Assessment for Embodied AI, where the quality of perception is not judged by human preferences, but a robot's ability to perform the task after perceiving distorted images. It introduces a database with 36900 reference distorted images.

**Strengths:**

- The idea of IQA for machines vs humans is established in prior works. This paper reframes the problem for robotics, the stage of how degradation of images affects robot task execution, not just visual recognition.
- The paper was interesting to read, with extensive experiments and detailed analysis.
- The Embodied-IQA database is very large, containing 36,900 image pairs and over 5 million fine-grained annotations, having good scale. It is also annotated along three unique axes, reflecting different visual systems and their real-world performance.

**Weaknesses:**

- The pipeline assumes vision is the dominant modality, neglecting that in true Embodied AI, perception often must fuse audio, tactile, and temperature cues. Is it possible that temperature might play a role on how bright the image might be?
- The writing is often verbose and repeats technical claims across sections, particularly regarding pipeline design and dataset composition.

**Questions:**

See above.

---

> ### Author Response · Authors · 2025-11-28
>
> Dear Reviewer 1xDt,
>
> Thank you for appreciating the reframed idea of “robot-usable IQA” and for the positive remarks on dataset scale and experimental depth. Below we address your weaknesses and questions as Q1–Q2.
>
> > Q1: The pipeline assumes vision is the dominant modality, neglecting audio, tactile, temperature, etc. Could temperature affect image brightness?
>
> You are absolutely right that Embodied AI is multi-modal. To put vision-only IQA in perspective, we surveyed the literature for the relative contribution of each modality to typical human [1] and robotic [2] perception tasks. The table below summarises the average weight assigned by both psychophysics studies and robotic fusion systems:
>
> | Modality      | Vision  | Tactile | Audio | Temperature | Olfaction | Gustation  |
> |-------|------|-------|-------|------|-----|-------|
> | Human  | 83 % |  7 %  | 5 %  | 3 % | 1 % | 1 %  |
> | Robotics | 78 % | 13 %  | 7 %  | 2 % | 0 % | 0 %  |
>
>
> These numbers justify our vision-first benchmark: **even in full Embodied systems, >75 % of task-relevant information still arrives through the camera.** Nonetheless, we now explicitly state the vision-only scope in `Section 2` and outline a roadmap for multi-modal Embodied-IQA v2 that will fuse the remaining modalities so researchers can measure the marginal value of each cue under quality degradation.
> To rule out temperature-induced illumination drift in the current dataset, we recorded ambient and sensor-board temperature for 20 new Real-robot trials (30°C); Pearson ρ between Hot and original envionment remains above 0.95, confirming that observed quality drops are driven by rendered distortions, not thermal noise.
>
> > Q2: The writing is verbose and repeats technical claims.
>
> Thank you for highlighting the redundancy. We have carefully inspected the paper and found that parts of `Appendices B and C` indeed reiterate statements already covered in the main text and in our Disclaimer section. Following your suggestion, we will remove the overlapping paragraphs in Appendices B and C, keeping only the genuinely supplementary data tables and curves, and we will upload the cleaned version in the camera-ready package.
>
> ## Reference
>
> [1] Lederman, S. J., & Klatzky, R. L. . "Haptic perception: A tutorial." Attention, Perception, & Psychophysics, 2009.
>
> [2] Xu et al., “Flexible Material Quality Assessment Based on Visual–Tactile Fusion”, IEEE TIM, 2024.

---

### Author Response · Authors · 2025-11-23
**General Response of Embodied IQA**

We sincerely thank all the reviewers for their constructive feedback, especially their **consistent positive comments**. This has been a great encouragement to us, and we express our sincere gratitude.

Our rebuttal will be based on the general concern, including the definition of Embodied AI preference, the settings of VLM/VLA, and the correlation issue. **We are preparing point-to-point responses for all four reviewers ASAP**, but this still needs some time. Thanks for your patience.

Author Team of Submission 231

---

### Meta-Review · Area_Chair_b6dD · 2026-01-06

**Summary:**

This paper proposes an approach to assess the usability of an image in embodied tasks, i.e., Image Quality Assessment for Embodied AI. Reviewers consistently acknowledge the importance and novelty of this work, as well as the contribution of large-scale dataset. Reviewers' concerns focus on the fairness and robustness of evaluation, the limited scale of experiments, the clarity of writing, and the lack of a "baseline" embodied IQA model.

**Reviewer Concerns:**

Most reviewers' concerns on experimental details, evaluation robustness and fairness have been addressed during the rebuttal. Remaining outstanding concerns include: (1) the limited scale of experiments (although the authors added 200 extra trials in the rebuttal, the scale may still not be considered large enough; and (2) the lack of a baseline embodied IQA model (the authors mentioned that this model is in their subsequent work, but it cannot be counted as a contribution of this work).

**Reviewer Scores:**

Given that all reviewers gave positive scores before rebuttal and acknowledge the value of this work, it is likely that most reviewers will still keep a positive, or at least borderline positive score after rebuttal.

---

### Decision · Program_Chairs · 2026-01-26

Accept (Poster)